# CERTIFIED ADVERSARIAL ROBUSTNESS UNDER THE BOUNDED SUPPORT SET

## ABSTRACT

Deep neural networks (DNNs) have revealed severe vulnerability to adversarial perturbations, besides empirical adversarial training for robustness, the design of provably robust classifiers attracts more and more attention. Randomized smoothing method provides the certified robustness with agnostic architecture, which is further extended to a provable robustness framework using $f$-divergence. While these methods cannot be applied to smoothing measures with bounded support sets such as uniform probability measures due to the use of likelihood ratio in their certification methods. In this paper, we introduce a framework that is able to deal with robustness properties of arbitrary smoothing measures including those with bounded support set, by using Wasserstein distance as well as total variation distance. By applying our methodology to uniform probability measures with support set $B_2(O, r)$, we obtain certified robustness properties concerning $l_p$-perturbations. And by applying to uniform probability measures with support set $B_\infty(O, r)$, we obtain certified robustness properties with respect to $l_1, l_2, l_\infty$-perturbations. We present experimental results on CIFAR-10 dataset with ResNet to validate our theory. It is worth mentioning that our certification procedure only costs constant computation time, which is an improvement upon the state-of-the-art methods in terms of the computation time.

## 1 INTRODUCTION

Vulnerability to adversarial samples is a major obstacle that various classifiers obtained by machine learning algorithms, especially deep neural networks (DNNs), need to overcome (Szegedy et al., 2013; Nguyen et al., 2015). For instance, in computer vision applications, deliberately adding some subtle perturbation $\delta$ that humans cannot perceive to the input image $x$ will cause DNNs to give a wrong classification output with high probability. Many empirical adversarial defenses have been proposed, among which adversarial training (Madry et al., 2018) is the most effective one (Athalye et al., 2018), however, it still faces stronger or adaptive attacks to decrease its effectiveness to a certain degree (Croce & Hein, 2020). This motivates research on certified robustness: algorithms that are provably robust to the worst-case attacks.

Some works propose algorithms to learn DNNs that are provably robust against norm-bounded adversarial perturbations by using some convex relaxation methods (Wong & Kolter, 2018; Weng et al., 2018; Xiao et al., 2018; Zhang et al., 2019). However, these approaches are usually computationally expensive and require extensive knowledge of classifier architecture. Besides, randomized smoothing (first introduced in Cohen et al. (2019)) has received significant attention in recent years for verifying the robustness of classifiers. Based on this method, several papers have studied which smoothing strategies perform better for specific $l_p$ perturbations. Cohen et al. (2019) concludes that randomized smoothing can be well understood for the $l_2$ case by using Gaussian probability measure for smoothing. It follows that several special cases of the conjecture have been proven for $p < 2$. Li et al. (2018) show that $l_1$ robustness can be achieved with the Laplacian distribution, and Lee et al. (2019) show that $l_0$ robustness can be achieved with a discrete distribution. Other papers start from the opposite perspective and focus on studying under specific assumptions which perturbation is provably difficult to handle and which smoothing methods are ineffective for particular disturbance. As for the existence of a noise distribution that works for the case of $p > 2$, Blum et al. (2020) and Kumar et al. (2020) show hardness results for random smoothing to achieve certified adversarial robustness against attacks in the $l_p$ ball of radius $\epsilon$. Nevertheless, since these works provide hardness results for every possible base classifier $f : \mathbb{R}^d \to \mathcal{Y}$ including those unusual and even bizarre ones, hardness results given by these papers might be attributed to taking into account classifiers that

will never appear in real-world applications. From this perspective, the order of difficulty restricted within the common classifiers subset still remains unresolved.

Notably, based on randomized smoothing strategy, Dvijotham et al. (2020) introduce a provable robustness framework using $f$-divergence as their convex relaxation technique. However, due to the use of likelihood ratio in their certification methods, the framework cannot be applied to smoothing measures with bounded support sets such as uniform probability measures. In this paper, we introduce a framework that is able to deal with robustness properties of arbitrary smoothing measures, including those with bounded support set, by using Wasserstein distance as well as total variation distance. Our contributions are summarized as follows:

- By applying our methodology to uniform probability measures with support set $B_2(O, r)$, we obtain certified robustness properties with respect to $l_p$-perturbations. By applying our methodology to uniform probability measures with support set $B_\infty(O, r)$, we obtain certified robustness properties with respect to $l_1, l_2, l_\infty$-perturbations.

- Furthermore, we analyze the cases when smoothing measure is taken as uniform probability measure with more general support set $B_p(O, r)$ and show the unavoidable curse of dimension for the usage of bounded support set smoothing measures.

- We present experimental results on CIFAR-10 dataset with ResNet to validate our theory. It is worth mentioning that our certification procedure only costs constant computation time, which is an improvement upon the state-of-the-art methods in terms of the computation time.

## 2 PROBLEM SETTING

Given a binary base classifier $h : \mathbb{R}^d \to \mathcal{Y} = \{\pm 1\}$ and smoothing probability measure $\mu$, the randomly smoothed classifier $h_\mu(x)$ is defined as follows.

**Definition 1** (smoothed classifier, smoothing measure). *The smoothed version of a base binary classifier $h$ producing labels in set $\mathcal{Y} = \{\pm 1\}$ is defined as*

$$h_\mu(x) = \arg\max_{y \in \mathcal{Y}} \mathbb{P}_{X \sim x + \mu}[h(X) = y], \tag{1}$$

*where $\mu \in \mathcal{P}(\mathcal{X})$ is called smoothing measure.*

Another way to understand this definition is to say that the smoothed classifier first scores point $x$ as $h_{\mu,y}(x) = \mathbb{P}_{X \sim x + \mu}[h(X) = y]$ for each specific class $y \in \mathcal{Y}$ and then outputs the class $y^*$ with the highest score. We want to study the robustness of the smoothed classifier $h_\mu$ against adversarial perturbations of size at most $\epsilon$ with respect to a given norm $||\cdot||_p$. The question that whether a bounded $l_p$ norm adversarial attack on a fixed input $x$ satisfying $h_\mu(x) = +1$ is successful or not can be formulated as solving the optimization problem below:

$$\min_{||x'-x||_p \leq \epsilon} \mathbb{P}_{X \sim x' + \mu}[h(X) = +1]. \tag{2}$$

The attack is successful if and only if the minimum value is smaller than $\frac{1}{2}$. Since we know little about the information of the black-box classifier $h$, we follow the approach introduced in Dvijotham et al. (2020): rather than studying the adversarial attack in the input space $\mathcal{X}$, we study it in the space of probability measures defined on input space $\mathcal{P}(\mathcal{X})$,

$$\min_{||x'-x|| \leq \epsilon} \mathbb{P}_{X \sim x' + \mu}[h(X) = +1] = \min_{\nu \in \mathcal{D}_{x,\epsilon,q}} \mathbb{P}_{X \sim \nu}[h(X) = +1], \tag{3}$$

where $\mathcal{D}_{x,\epsilon,q} := \{x' + \mu : ||x - x'||_q \leq \epsilon\}$ represents an $l_q$-norm-based constraint set of radius $\epsilon$ for smoothing measure $\mu$ centered at a particular sample point $x$. Then, we follow the full-information robust certification framework established in Dvijotham et al. (2020) and analyze the generalization of binary classifier $h$, which they called specification and denote it as $\phi : \mathcal{X} \subseteq \mathbb{R}^d \to \mathcal{Z} \subseteq \mathbb{R}$. Besides, define reference measure $\rho$ as $x + \mu$ and a collection of perturbed probability measures $\mathcal{D} \subseteq \mathcal{P}(\mathcal{X})$. Checking whether a given specification $\phi$ is robustly certified at $\rho$ with respect to $\mathcal{D}$ or not is equivalent to estimating the optimal value of following optimization problem is non-negative or not:

$$OPT(\phi, \rho, \mathcal{D}) := \min_{\nu \in \mathcal{D}} \mathbb{E}_{X \sim \nu}[\phi(X)]. \tag{4}$$

And certifying $l_p$ robustness on input $x$ with output of smoothed classifier $h_\mu(x) = +1$ is equivalent to verify whether $OPT(h, x + \mu, \mathcal{D}_{x,\epsilon,q}) \geq 0$ or not.

## 3 OUR CERTIFICATION PROCEDURES

Notice that the certification in Dvijotham et al. (2020) uses the likelihood ratio $r(X) = \frac{\nu(X)}{\rho(X)}$, while $r(X)$ is well-defined only when the support set of $\rho$ contains the support set of $\nu$. Thus, when the support set of reference measure $\rho$ is bounded, and $\nu$ takes even a small translation of $\rho$, the support set of $\nu$ will cross over the boundary of support set of $\rho$. Their certification is invalid in this case. In this paper, by using Wasserstein distance as well as total variance distance, we provide analytical techniques able to analyze bounded support set which is not covered by Dvijotham et al. (2020).

Since the set of measures $\mathcal{D}_{x,\epsilon,q}$ constraint in optimization problem $OPT(h, x + \mu, \mathcal{D}_{x,\epsilon,q})$ is intractable to deal with, we consider relaxations of this by using Wasserstein distance as well as total variance distance constraints between $\nu$ and $x + \mu$, i.e. $\mathcal{D}_{x,\epsilon,q} \subseteq \{\nu : W_p(x + \mu, \nu) \leq \delta\} := \mathcal{D}_{x,\delta,p}$ which represents $W_p$-distance-based constraint set of radius $\delta$ for smoothing measure $\mu$ centered at sample point $x$ and $\mathcal{D} \subseteq \{\nu : TV(x + \mu, \nu) \leq \xi\} := \mathcal{D}_{x,\xi}$ which represents TV-distance-based constraint set of radius $\xi$ for smoothing measure $\mu$ centered at sample point $x$. Combining the two relaxations, we know $\mathcal{D}_{x,\epsilon,q} \subseteq \mathcal{D}_{x,\delta,p} \cap \mathcal{D}_{x,\xi}$ and therefore

$$OPT(h, x + \mu, \mathcal{D}_{x,\epsilon,q}) \geq OPT(h, x + \mu, \mathcal{D}_{x,\delta,p} \cap \mathcal{D}_{x,\xi}). \tag{5}$$

And we can obtain tighter relaxation by combining multiple Wasserstein distance relaxation with different $p$, i.e. $\mathcal{D}_{x,\epsilon,q} \subseteq \left( \bigcap_{i \in \mathcal{I}} \mathcal{D}_{x,\delta_i,p_i} \right) \cap \mathcal{D}_{x,\xi}$ where $\mathcal{I} \subseteq \mathbb{R}^+$ and therefore

$$OPT(h, x + \mu, \mathcal{D}_{x,\epsilon,q}) \geq OPT\left(h, x + \mu, \left( \bigcap_{i \in \mathcal{I}} \mathcal{D}_{x,\delta_i,p_i} \right) \cap \mathcal{D}_{x,\xi}\right). \tag{6}$$

Thus, for a fixed input $x$, it suffices to consider the Wasserstein distance and total variance distance relaxed problem and verify whether $OPT(h, x + \mu, \mathcal{D}_{x,\delta,p} \cap \mathcal{D}_{x,\xi}) \geq 0$ or not. The analysis of this problem can be divided into three parts: computing the Wasserstein distance relaxation measure set, computing the total variance distance relaxation measure set, and computing the Lagrange function as well as dual problem of the relaxed optimization problem $OPT(h, x + \mu, \mathcal{D}_{x,\delta,p} \cap \mathcal{D}_{x,\xi})$. The details are discussed in the following three sections.

### 3.1 RELAXATION USING WASSERSTEIN DISTANCE

In this section, we show the following relaxation from norm-based constraint sets into W-distance-based constraint sets for general smoothing measures as well as Gaussian smoothing measure.

$$\mathcal{D}_{x,\epsilon,q} = \{x' + \mu : ||x - x'||_q \leq \epsilon\} \subseteq \{\nu : W_p(x + \mu, \nu) \leq \delta\} = \mathcal{D}_{x,\delta,p}. \tag{7}$$

#### 3.1.1 GENERAL PROBABILITY MEASURE

Here, we want to find a $\delta_q(\epsilon)$ such that

$$\mathcal{D}_{x,\epsilon,q} = \{x' + \mu : ||x - x'||_q \leq \epsilon\} \subseteq \{\nu : W_p(x + \mu, \nu) \leq \delta_q(\epsilon)\} = \mathcal{D}_{x,\delta_q(\epsilon),p} \text{ for all } \mu \in \mathcal{P}(\mathcal{X}). \tag{8}$$

**Theorem 3.1.** *For all $x \in \mathbb{R}^d, \epsilon > 0, q > 0$, norm-based constraint set $\mathcal{D}_{x,\epsilon,q}$ can be relaxed into W-distance-based constraint set $\mathcal{D}_{x,\delta_q(\epsilon),p}$ with radius $\delta_q(\epsilon) = \max\{\epsilon, \epsilon d^{\frac{1}{2} - \frac{1}{q}}\}$ which can be formulated as*

$$\mathcal{D}_{x,\epsilon,q} \subseteq \left\{\nu : W_p(x + \mu, \nu) \leq \max\{\epsilon, \epsilon d^{\frac{1}{2} - \frac{1}{q}}\}\right\} := \mathcal{D}_{x,\max\{\epsilon, \epsilon d^{\frac{1}{2} - \frac{1}{q}}\},p} \tag{9}$$

*And this relaxation radius $\max\{\epsilon, \epsilon d^{\frac{1}{2} - \frac{1}{q}}\}$ works for any Wasserstein distance parameter $p > 0$ as well as any smoothing measure $\mu$.*

Note that for $l_q(q \leq 2)$ adversarial perturbations, the relaxed radius avoids the influence of dimension $d$, whereas for $l_q(q > 2)$ adversarial perturbations, as $q$ increases, $\frac{1}{2} - \frac{1}{q}$ increases from 0 to $\frac{1}{2}$ correspondingly. The fact that the radius of $W_q$-distance constraint set grows with order $\Theta(d^{\frac{1}{2} - \frac{1}{q}})$ provides us with an intuition that it is increasingly harder to bound $\mathcal{D}_{x,\epsilon,q}$ with larger $q$, therefore, W-distance-relaxation works better for $l_q(q \leq 2)$ norm perturbation. And this relaxation radius is tight for $W_2$ distance and Gaussian smoothing measures which is proved in the appendix D and therefore shows that $W_2$-distance-relaxation works well for Gaussian smoothing measure.

### 3.2 Relaxation Using Total Variance Distance

In this section, we show the following relaxation from norm-based constraint sets into TV-distance-based constraint sets for Gaussian and uniform smoothing measures.

$$\mathcal{D}_{x,\epsilon,q} = \{x' + \mu : ||x - x'||_q \leq \epsilon\} \subseteq \{\nu : TV(x + \mu, \nu) \leq \xi\} = \mathcal{D}_{x,\xi} \tag{10}$$

#### 3.2.1 Gaussian Probability Measure

Here, we want to find a $\xi(\epsilon)$ for Gaussian measure $\mu = \mathcal{N}(0, \sigma^2 I)$ such that

$$\mathcal{D}_{x,\epsilon,q} = \{x' + \mu : ||x - x'||_q \leq \epsilon\} \subseteq \{\nu : TV(x + \mu, \nu) \leq \xi(\epsilon)\} = \mathcal{D}_{x,\xi(\epsilon)}. \tag{11}$$

The magnitude of $\xi(\epsilon)$ is given by the following theorem.

**Theorem 3.2.** *For Gaussian probability measure $\mu = \mathcal{N}(0, \sigma^2 I)$ on Euclidean space $\mathbb{R}^d$ and for all $x \in \mathbb{R}, \epsilon > 0, q > 0$, norm-based constraint set $D_{x,\epsilon,q}$ can be relaxed into TV-distance-based constraint set $\mathcal{D}_{x,\xi(\epsilon)}$ with radius $\xi(\epsilon) = 2G\left(\frac{\max\{\epsilon, \epsilon d^{\frac{1}{2} - \frac{1}{q}}\}}{2\sigma}\right) - 1$ where $G$ is the cumulative distribution function for standard normal distribution $\mathcal{N}(0, 1)$ which can be formulated as*

$$D_{x,\epsilon,q} \subseteq \left\{\nu : TV(x + \mu, \nu) \leq 2G\left(\max\left\{\frac{\epsilon}{2\sigma}, \frac{\epsilon d^{\frac{1}{2} - \frac{1}{q}}}{2\sigma}\right\}\right) - 1\right\}. \tag{12}$$

This theorem theoretically shows that TV distance relaxation works effectively for $l_q(q \leq 2)$ perturbation due to the irrelevance of the radius to dimension $d$ and increasingly bad for $l_q(q > 2)$ perturbation because of the dependence of the radius to dimension $d$ as order $\Theta(d^{\frac{1}{2} - \frac{1}{q}})$.

#### 3.2.2 Uniform Probability Measure

Here, we want to find a $\xi(\epsilon)$ for uniform measure $\mu = \mathcal{U}(K)$, where $K$ is a specific convex compact set, with density function $f_K(x) = \frac{1}{\text{Vol}(K)}\mathcal{I}_{x \in K}$ such that

$$\mathcal{D}_{x,\epsilon,q} = \{x' + \mu : ||x - x'||_q \leq \epsilon\} \subseteq \{\nu : TV(x + \mu, \nu) \leq \xi(\epsilon)\} = \mathcal{D}_{x,\xi(\epsilon)}. \tag{13}$$

In this paper, we mainly focus on the case when $K$ is $l_p$-norm ball centered at original point $O$ with radius $r$, i.e., $K = B_p(O, r)$. We give following theorems about special cases when $p = 1, 2, \infty$.

**Theorem 3.3.** *When $K$ is an $l_1$ norm ball centered at $O$ with radius $r$, for uniform probability measure $\mathcal{U}(K)$ on Euclidean space $\mathbb{R}^d$, we have*

$$\mathcal{D}_{x,\epsilon,q} \setminus \{\nu : TV(x + \mu, \nu) \leq 1 - \delta\} \neq \emptyset \text{ for all } q > 1 \text{ and arbitrarily small } \delta > 0, \tag{14}$$

*when $\epsilon \geq 2rd^{\frac{1}{q} - 1}$. Note that $\epsilon \geq \frac{2r}{\sqrt{d}}$ which decays with order $\Theta(d^{-\frac{1}{2}})$ for $q = 2$ and $\epsilon \geq \frac{2r}{d}$ which decays with order $\Theta(d^{-1})$ for $q = \infty$.*

This theorem theoretically shows that for uniform smoothing measures with $l_1$ ball support set, and total variance distance failed to relax measure set $\mathcal{D}_{x,\epsilon,q}$ effectively when $q = 2, \infty$. And this will consequently lead to bad performance for $l_2$ and $l_\infty$ robustness certification task, which can be seen from the following section discussing the importance of TV-distance-based relaxation radius.

**Theorem 3.4.** *When $K$ is an $l_2$ (Euclidean) ball centered at $O$ with radius $r$, for uniform probability measure $\mathcal{U}(K)$ on Euclidean space $\mathbb{R}^d$ and for all $x \in \mathbb{R}, \epsilon > 0, q > 0$, when $\epsilon > \min\{2r, 2rd^{\frac{1}{q} - \frac{1}{2}}\}$, norm-based constraint set $\mathcal{D}_{x,\epsilon,q}$ failed to be relaxed into TV-distance-based constraint set which can be formulated as*

$$\mathcal{D}_{x,\epsilon,q} \setminus \{\nu : TV(x + \mu, \nu) \leq 1 - \delta\} \neq \emptyset \text{ for all } q > 1 \text{ and arbitrarily small } \delta > 0. \tag{15}$$

*And when $\epsilon \leq \min\{2r, 2rd^{\frac{1}{q} - \frac{1}{2}}\}$, norm-based constraint set $D_{x,\epsilon,q}$ can be relaxed into valid TV-distance-based constraint set $\mathcal{D}_{x,\xi(\epsilon)}$ with radius $\xi(\epsilon) = 1 - \frac{\int_0^{\arccos\left(\frac{\max\{\epsilon, \epsilon d^{\frac{1}{2} - \frac{1}{q}}\}}{2r}\right)} \sin^n(t)dt}{\int_0^{\frac{\pi}{2}} \sin^n(t)dt}$ which can be formulated as*

$$\mathcal{D}_{x,\epsilon,q} \subseteq \left\{\nu : TV(x + \mu, \nu) \leq 1 - \frac{\int_0^{\arccos\left(\frac{\max\{\epsilon, \epsilon d^{\frac{1}{2} - \frac{1}{q}}\}}{2r}\right)} \sin^n(t)dt}{\int_0^{\frac{\pi}{2}} \sin^n(t)dt}\right\}. \tag{16}$$

This theorem shows that for uniform smoothing measures with $l_2$ ball support set, when $q \leq 2$, relaxation radius is independent of dimension $d$, whereas when $q > 2$ relaxation radius starts to be bound up with dimension $d$ and the impact of $d$ grows as $q$ increases. To put it another way, total variance distance relaxation performs well for uniform smoothing measures with $l_2$ ball support set when $q \leq 2$ and increasingly poor when $q > 2$.

**Theorem 3.5.** *When $K$ is an $l_\infty$ cube centered at $O$ with radius $r$, for uniform probability measure $\mathcal{U}(K)$ on Euclidean space $\mathbb{R}^d$ and for all $x \in \mathbb{R}, \epsilon > 0, q > 0$, when $\epsilon > 2r$, norm-based constraint set $\mathcal{D}_{x,\epsilon,q}$ failed to be relaxed into TV-distance-based constraint set which can be formulated as*

$$\mathcal{D}_{x,\epsilon,q} \setminus \left\{ \nu : TV(x + \mu, \nu) \leq 1 - \delta \right\} \neq \emptyset \text{ for all } q > 0 \text{ and arbitrarily small } \delta > 0. \quad (17)$$

*And when $\epsilon \leq 2r$, norm-based constraint set $D_{x,\epsilon,q}$ can be relaxed into valid TV-distance-based constraint set $\mathcal{D}_{x,\xi(\epsilon)}$. When $q = 1$, $\xi(\epsilon)$ can be taken as $\frac{\epsilon}{2r}$, which can be formulated as*

$$D_{x,\epsilon,1} \subseteq \left\{ \nu : TV(x + \mu, \nu) \leq \frac{\epsilon}{2r} \right\}. \quad (18)$$

*When $q = 2$, $\xi(\epsilon)$ can be taken as $1 - \left(1 - \frac{\epsilon}{2d^{\frac{1}{2}}r}\right)^d$ when $0 < \epsilon \leq 2t_n r$ where $\sqrt{\frac{n-1}{n}} \leq t_n < 1$ and $t_n$ approaches $1$ at an exponential rate, which can be formulated as*

$$D_{x,\epsilon,2} \subseteq \left\{ \nu : TV(x + \mu, \nu) \leq 1 - \left(1 - \frac{\epsilon}{2d^{\frac{1}{2}}r}\right)^d \right\} \approx \left\{ \nu : TV(x + \mu, \nu) \leq 1 - e^{-\frac{\epsilon}{2r}d^{\frac{1}{2}}} \right\}, \quad (19)$$

*$\xi(\epsilon)$ can be taken as $1 - \left(\frac{d-1+\sqrt{d(\frac{\epsilon}{2r})^2 - d + 1}}{d}\right)^{d-1}\left(\frac{1 - \sqrt{d(\frac{\epsilon}{2r})^2 - d + 1}}{d}\right)$ when $2t_n r < \epsilon < 2r$, which can be formulated as*

$$\mathcal{D}_{x,\epsilon,2} \subseteq \left\{ \nu : TV(x + \mu, \nu) \leq 1 - \left(\frac{d - 1 + \sqrt{d(\frac{\epsilon}{2r})^2 - d + 1}}{d}\right)^{d-1}\left(\frac{1 - \sqrt{d(\frac{\epsilon}{2r})^2 - d + 1}}{d}\right) \right\}. \quad (20)$$

*When $q = \infty$, $\xi(\epsilon)$ can be taken as $1 - \left(1 - \frac{\epsilon}{2r}\right)^d$, which can be formulated as*

$$\mathcal{D}_{x,\epsilon,\infty} \subseteq \left\{ \nu : TV(x + \mu, \nu) \leq 1 - \left(1 - \frac{\epsilon}{2r}\right)^d \right\}. \quad (21)$$

As for uniform smoothing measures with $l_\infty$ cube support set, this theorem shows that the performance towards $l_1$ perturbation turns out to be fine since TV distance relaxation radius $\frac{\epsilon}{2r}$ has nothing to do with dimension $d$ and the dimensional curse is avoided. However, in this case, TV distance relaxation shows incapability to cope with $l_2$ and $l_\infty$ perturbation in some extent due to the rate of increasing radius tending to 1 as $\Theta(e^{d^{\frac{1}{2}}})$ and $\Theta(e^d)$.

After discussing the special cases when $K$ is an $l_\infty$ cube or an $l_2$ Euclidean ball, we then consider the general case when $K$ is an $l_p$ ball centered at the original point with radius $r$ and give a lower bound for TV distance relaxation radius in the following theorem.

**Theorem 3.6.** *When $K$ is an $l_p$ ball centered at $O$ with radius $r$, for uniform probability measure $\mathcal{U}(K)$ on Euclidean space $\mathbb{R}^d$ and assume norm-based constraint set $\mathcal{D}_{x,\epsilon,q}$ can be relaxed into TV-distance-based constraint set $\mathcal{D}_{x,\xi(\epsilon)}$, then*

$$\xi(\epsilon) \geq 2 \int_0^{\frac{\epsilon d^{\frac{1}{p}}}{4r(pe)^{\frac{1}{p}}\Gamma(1+\frac{1}{p})}} \exp\left(\frac{1}{p} - e(2x\Gamma(1 + \frac{1}{p}))^p\right) dx$$

*for all perturbation norm parameter $q > 0$ with high probability when $d$ is sufficiently large, which can be formulated as*

$$\mathcal{D}_{x,\epsilon,q} \setminus \left\{ \nu : TV(x + \mu, \nu) \leq 2 \int_0^{\frac{\epsilon d^{\frac{1}{p}}}{4r(pe)^{\frac{1}{p}}\Gamma(1+\frac{1}{p})}} \exp\left(\frac{1}{p} - e(2x\Gamma(1 + \frac{1}{p}))^p\right) dx - \delta \right\} \neq \emptyset \quad (22)$$

*for arbitrarily small $\delta > 0$.*

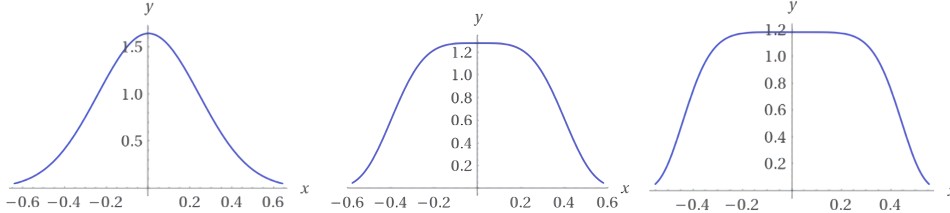

Figure 1: Graph of density function $f(x) = \exp\left(\frac{1}{p} - e(\frac{2}{p}\Gamma(\frac{1}{p}))^p x^p\right)$ when $p = 2, 4, 6$ from left to right

A way to interpret this theorem is that as $p$ increases and $K$ correspondingly translates from $l_1$ norm cross-polytope into $l_\infty$ norm cube, the dependence of integral upper limit $\frac{\epsilon d^{\frac{1}{p}}}{4r(pe)^{\frac{1}{p}}\Gamma(1+\frac{1}{p})}$ on dimension $d$ is gradually reduced, which theoretically shows by taking all kinds of $l_q$ perturbations into consideration, in average scale $\mathcal{U}(B_p(O, r))$ tends to perform better than $\mathcal{U}(B_q(O, r))$ where $p > q$. Nevertheless, note that the curse of dimension is unavoidable when we use uniform smoothing measure $\mathcal{U}(K)$ with bounded support set. Note the graphs of density function shown in Figure 1 when $p = 2, 4, 6$, we know that as $p$ increases, the density curve is increasingly short, fat, and light-tailed. From another perspective, we consider a ball with a fixed radius $r$. As the dimension $d$ of the base Euclidean space increases, fixed proportion of mass concentrates within a slab of width $\Theta(d^{-\frac{1}{p}})$. Thus, intuitively, it is increasingly difficult to bound the perturbed measure set $\mathcal{D}_{x,\epsilon,q}$ by using TV distance and certify as dimension $d$ enlarge due to the existence of TV-relaxation upper bound in the dual optimization problem.

### 3.3 VERIFYING FULL-INFORMATION ROBUST CERTIFICATION

Based on the above analysis, in this section, we are now prepared to compute the Lagrange function and dual problem of the relaxed optimization problem $OPT(\phi, x + \mu, \mathcal{D}_{x,\delta,p} \cap \mathcal{D}_{x,\xi})$. Here we mainly focus on the case when reference measure $\rho = x + \mu$ and perturbed probability measure $\nu$ are absolutely continuous w.r.t. Lebesgue measure $\lambda$ on $\mathbb{R}^d$, i.e., $\rho, \nu \ll \lambda$ and discard uncommon cases when $\rho, \nu$ are discrete, single or mixed w.r.t. $\lambda$. Since $\rho, \nu \ll \lambda$, assume the density function of $\rho$ and $\nu$ w.r.t. Lebesgue measure $\lambda$ are $f(x)$ and $g(x), x \in \mathbb{R}^d$ respectively. Instead of using likelihood ratio $r(x)$, we consider the difference between $g(x)$ and $f(x)$ and define it as $q(x) := g(x) - f(x)$. The objective function $\mathbb{E}_{X \sim \nu}[\phi(X)]$ of optimization problem $OPT(\phi, \rho, \mathcal{D})$ can be rewritten in terms of difference function $q(x)$. And we give the theorems below.

**Theorem 3.7** ($W_p$ distance relaxation with $0 < p \leq 1$). *The relaxed optimization problem $OPT(\phi, x + \mu, \mathcal{D}_{x,\delta,p} \cap \mathcal{D}_{x,\xi})$ is equivalent to the convex optimization problem with only one functional variable as below*

$$\inf_{q \in \mathcal{L}^1(\mathcal{X})} \int_{\mathcal{X}} \phi(x)q(x)dx + \mathbb{E}_{X \sim x+\mu}[\phi(X)], \ s.t. \ \sup_{||f||_{L,p} \leq 1} \int f(x)q(x)dx \leq \delta, \int |q(x)|dx \leq 2\xi \tag{23}$$

*where* $||f||_{L,p} := \sup_{x,y \in \mathbb{R}^d, x \neq y} \frac{|f(x)-f(y)|}{||x-y||_2^p}$.

**Theorem 3.8** ($W_p$ distance relaxation with $p > 1$). *When smoothing measure $\mu$ possesses a convex compact support set $K$ and $R := \sup_{y \in K} ||y||_2$, $R^* := ||x||_2 + R + \max\{\epsilon, \epsilon d^{\frac{1}{2} - \frac{1}{q}}\}$, the relaxed optimization problem $OPT(\phi, x+\mu, \mathcal{D}_{x,\delta,p} \cap \mathcal{D}_{x,\xi})$ can be further relaxed into the convex optimization problem with only one functional variable below*

$$\inf_{q \in \mathcal{L}^1(\mathcal{X})} \int_{\mathcal{X}} \phi(x)q(x)dx + \mathbb{E}_{X \sim x+\mu}[\phi(X)], \tag{24}$$

$$s.t. \sup_{||f||_L \leq p(2R^*)^{p-1}} \int f(x)q(x)dx \leq \delta^p + (p-1)(2R^*)^{p-1}, \int |q(x)|dx \leq 2\xi,$$

*where* $||f||_L := \sup_{x,y \in \mathbb{R}^d, x \neq y} \frac{|f(x)-f(y)|}{||x-y||_2}$.

**Theorem 3.9.** *The Lagrange function of optimization problem in* (23) *and* (24) *is*

$$L(\lambda) = \mathbb{E}_{X \sim x+\mu}[\phi(X)] - 2\xi - \lambda C, \tag{25}$$

| Smoothing Measure | Perturbation | Certification Objective | Prerequisite |
|---|---|---|---|
| $\mathcal{U}(B_2(O,r))$ | $l_q(q\le 2)$ | $\mathbb{E}_{X\sim x+\mu}[\phi(X)]-2\left(1-\dfrac{\int_0^{\arccos(\frac{\epsilon}{2r})}\sin^n(t)dt}{\int_0^{\frac{\pi}{2}}\sin^n(t)dt}\right)$ | $\epsilon\le 2r$ |
| | $l_q(q>2)$ | $\mathbb{E}_{X\sim x+\mu}[\phi(X)]-2\left(1-\dfrac{\int_0^{\arccos(\frac{\epsilon d^{\frac{1}{2}-\frac{1}{q}}}{2r})}\sin^n(t)dt}{\int_0^{\frac{\pi}{2}}\sin^n(t)dt}\right)$ | $\epsilon\le 2rd^{\frac{1}{q}-\frac{1}{2}}$ |
| $\mathcal{U}(B_\infty(O,r))$ | $l_1$ | $\mathbb{E}_{X\sim x+\mu}[\phi(X)]-\frac{\epsilon}{r}$ | $\epsilon\le 2r$ |
| | $l_2$ | $\mathbb{E}_{X\sim x+\mu}[\phi(X)]-2\left(1-\left(1-\frac{\epsilon}{2d^{\frac{1}{2}}r}\right)^d\right)$ | $\epsilon\le 2t_n r$ |
| | $l_\infty$ | $\mathbb{E}_{X\sim x+\mu}[\phi(X)]-2\left(1-\left(1-\frac{\epsilon}{2r}\right)^d\right)$ | $\epsilon\le 2r$ |
| $\mathcal{N}(0,\sigma^2 I)$ | $l_q(q\le 2)$ | $\mathbb{E}_{X\sim x+\mu}[\phi(X)]-2\left(2G(\frac{\epsilon}{2\sigma})-1\right)$ | - |
| | $l_q(q>2)$ | $\mathbb{E}_{X\sim x+\mu}[\phi(X)]-2\left(2G(\frac{\epsilon d^{\frac{1}{2}-\frac{1}{q}}}{2\sigma})-1\right)$ | - |

Table 1: Certification objectives and prerequisites.

*where $\lambda \ge 0$ is the dual variable w.r.t. constraint $\sup_{||f||_L\le 1}\int f(x)q(x)dx \le \delta$ or constraint* $\sup_{||f||_L\le p(2R^*)^{p-1}}\int f(x)q(x)dx \le \delta^p+(p-1)(2R^*)^{p-1}$ *and $C:=\delta$ when $0<p\le 1$ whereas* $C:=\delta^p+(p-1)(2R^*)^{p-1}$ *when $p>1$.*

Using the duality result, we know the optimal value in (23) can be obtained by computing

$$\max_{\lambda\ge 0}\mathbb{E}_{X\sim x+\mu}[\phi(X)]-\xi-\lambda C = \mathbb{E}_{X\sim x+\mu}[\phi(X)]-\xi, \tag{26}$$

which is only related to the radius $\xi$ of TV distance relaxation set. We can see from this formula the significance of TV distance relaxation radius. By plugging the TV distance relaxation radius given in theorem 3.4, 3.5 and 3.2 in dual optimization problem, we obtain the certification objective in Table 1 and we return certified for $l_p$ norm perturbation with magnitude $\epsilon$ if the objective function has non-negative value.

### 3.4 RELATIONSHIP WITH PREVIOUS WORK

By applying our methodology to Gaussian probability measure, we miraculously obtain the same certified robustness properties provided in Dvijotham et al. (2020) using as Hockey-stick divergence with $\beta=1$.

**Theorem 3.10.** *When smoothing measure is taken as Gaussian probability measure, the certificate* $\mathbb{E}_{X\sim x+\mu}[\phi(X)]-2\left(2G(\frac{\epsilon}{2\sigma})-1\right)$ *given in our paper is equivalent to the certificate $\epsilon_{HS,1}\le[\frac{\theta_a-\theta_b}{2}]_+$ given in paper Dvijotham et al. (2020).*

Therefore, when applying both methodologies to Gaussian measure, the formulas obtained are theoretically equivalent. Despite the similarity in analyzing Gaussian measure, our work covers cases with bounded support sets, which is our main contribution.

## 4 EXPERIMENTS

For adversarial robustness certification, we choose the *test set certified accuracy* as our metric of interest, which is defined as the percentage of data points in the test set that can be correctly classified and can also pass the robustness certification within an $l_2$ ball of an assigned radius $r$. To pass the robustness certification at data point $x$, the classification results of all points within an $l_2$ distance to the original point $x$ must be consistent. For a model using the smoothing method, the classification result of a data point is the class with the highest score in the smoothed data distribution, not to be confused with the direct output of the base classifier at that data point.

In all experiments, the certification process on the test set with assigned perturbation $l_2$ radius is shown in the following Algorithm 1. Note that the $cert(score_a, 1-score_a, r)$ function returns true if the certification objective is non-negative, otherwise it returns false. Such objective is calculated

using formulas in Table 1 with $l_2$ perturbation and corresponding smoothing distribution. Since our method using Wasserstein distance does not involve iterations, our certification procedure only costs constant computation time, which is much faster than Dvijotham et al. (2020).

---

**Algorithm 1** Certification process

---

**Input:** $T$: test set, $gt(x)$: true class of image $x$, $f(x)$: base classifier, $D(x)$: smoothing distribution, $n$: sample amount, $r$: perturbation radius
**Output:** $acc$: test set certified accuracy
1: $certCount \leftarrow 0, allCount \leftarrow 0$
2: **for all** $x \in T$ **do**
3:     $S \leftarrow \{n \text{ samples from } D(x)\}$
4:     $count_c \leftarrow 0$ for every class $c$
5:     **for all** $x' \in S$ **do**
6:         $count_{f(x')} \leftarrow count_{f(x')} + 1$
7:     **end for**
8:     $score_c \leftarrow \frac{count_c}{card(S)}$ for every class $c$
9:     $a \leftarrow \arg\max_c \{score_c\}$
10:     **if** $a = gt(x) \wedge cert(score_a, 1 - score_a, r)$ **then**
11:         $certCount \leftarrow certCount + 1$
12:     **end if**
13:     $allCount \leftarrow allCount + 1$
14: **end for**
15: **return** $acc \leftarrow \frac{certCount}{allCount}$

---

We achieve identical results when comparing our W-distance method with the F-divergence method in Dvijotham et al. (2020) using Gaussian distribution and with specific metric parameter settings, which is proved possible in Section 3.4. However, there is no previous work done yet to examine the usage of uniform distribution when smoothing, so we mainly focus on comparing Gaussian, $l_2$ and $l_\infty$ uniform distribution all using our W-distance method.

### 4.1 SETUPS

We choose CIFAR-10 as our dataset and ResNet-110 as our base classifier. We firstly train the base classifier on the 50000 image training set without smoothing and achieve 89.6% prediction accuracy on the 10000 image test set. Then we run the certification process on the test set with incremental perturbation radius $r$. We test out different smoothing distributions as mentioned above, and we change the parameters of such distributions to illustrate the effect of different distributions further. We also try increasing the smoothing sample amount to examine the trade-off between performance and accuracy improvement. All training, testing, and certification are run on an NVIDIA RTX 3090.

### 4.2 W-DISTANCE METHOD WITH DIFFERENT SMOOTHING DISTRIBUTIONS

We firstly implement our W-distance method with $N(x, 0.05)$ as smoothing distribution and sample amount $n$=100, and then we change the variance of the Gaussian distribution to 0.025 and 0.1. As shown in Figure 2, there is a neat cut-off for each setting where the perturbation gets too big, and no data point can pass the certification at this point. By changing the variance of the smoothing distribution, we observe a clear trend that the increase of variance leads to a drop of initial certification

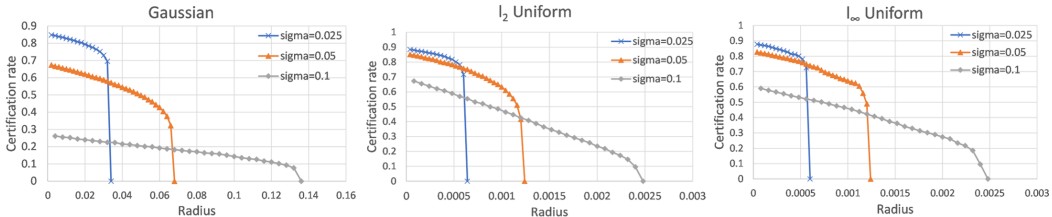

Figure 2: Results of different smoothing distributions using our W-distance method. Sigma stands for variance for Gaussian distribution and the norm range for uniform distribution.

accuracy but also stronger robustness that can endure more significant perturbation, and the decrease of the variance leads to the opposite change accordingly.

Next, for smoothing process, we substitute Gaussian distribution with $l_2$ or $l_\infty$ uniform distribution, with the norm range set to 0.025, 0.05 and 0.1. In Figure 2, both experiment results show identical characteristics as with Gaussian distribution, but they bring along a critical issue, the mismatch of the perturbation radius's magnitude. Comparing the perturbation radius at the cut-off point, we find that the radius of Gaussian distribution is about 50 times larger than that of two uniform distributions.

We assume that this phenomenon is caused by the lack of intersection of the smoothing distributions before and after perturbation. For Gaussian distribution, there is always an intersection no matter how big the perturbation radius gets, but two uniform distributions will separate quickly and become disjoint under perturbation. Furthermore, the dimension of a $32 \times 32 \times 3$ image is 3072, the square root of which is around 55.4, very close to the cut-off radius's 50 times ratio difference. Such correlation may trace to the involvement of dimension when calculating the finite support set volume of $l_2$ and $l_\infty$ uniform distribution, while the support set volume of Gaussian distribution is infinitely large. We conjecture that such deficiency is inherent when using the uniform distribution, which can hardly be further improved.

### 4.3 W-DISTANCE METHOD WITH DIFFERENT SAMPLING AMOUNTS

When calculating scores for each class in the smoothing process, as we cannot classify all possible data points, we shall only acquire approximate scores by sampling from the smoothed data distribution. Thus such scores may differ in multiple runs due to the randomness of sampling. However, through our experiments, we find that with a certain amount of samples, we can already obtain sufficiently accurate scores, which cannot be significantly improved by increasing the sample amount.

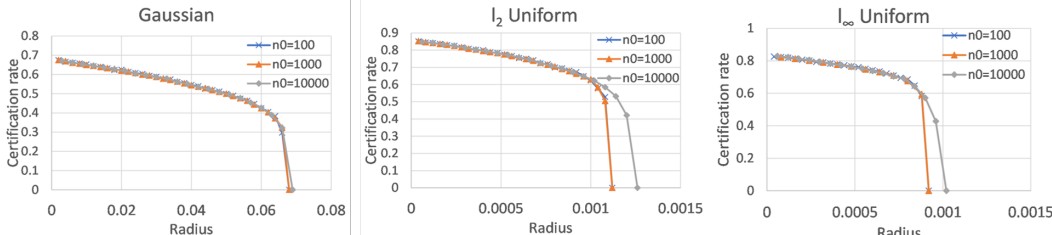

Figure 3: Results on sample amounts with different smoothing distributions using W-distance.

We set the sample amount $n$ to 100, 1000, and 10000 with three different smoothing distributions, and they all obtain similar results: it takes only 10 minutes to run through the 10000 images test set with 100 samples for each image, 30 minutes with 1000 samples and 3 hours with excessive 10000 samples. It is ten times faster than the 2 hours running time with the iteration-based method in Dvijotham et al. (2020) using just 100 samples. It is also worth noting in Figure 3 that by increasing the sample amount, no significant improvement is observed with Gaussian distribution. However, there is minor progress made with both uniform distributions when the samples are getting overly abundant. We assume that the extra samples make up for the lack of intersections of smoothing uniform distributions before and after the perturbation, while Gaussian distribution has no such issues.

## 5 CONCLUSION

We have introduced a framework based on Wasserstein distance and total variance distance relaxation as well as Lagrange duality. This methodology is able to deal with the analysis of bounded support set smoothing measures, which is not covered by previous work. Moreover, we have analyzed the performance of specific smoothing measures, including Gaussian probability measure and uniform probability measures with support set $B_2(O, r), B_\infty(O, r)$ theoretically and experimentally, which shows the relative incapability of bounded support set smoothing measures compared with Gaussian smoothing measure.

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

## A    OPTIMAL TRANSPORT THEORY

Assume $\mu, \nu \in \mathcal{P}(\mathbb{R}^d)$. Besides, assume $\mu, \nu$ are absolutely continuous w.r.t. Lebesgue measure $\lambda$ and let density functions be $f$ and $g$.

**Definition 2** (Push Forward). *If $T : \mathbb{R}^d \to \mathbb{R}^d$, then the distribution of $T(X)$ is called the push-forward of $P$, denoted by $T_{\#}P$. In other words,*

$$T_{\#}P(A) = P(T(x) \in A) = P(T^{-1}(A))$$

**Definition 3** (Optimal Distance, Optimal Transport Map). *The Monge version of the optimal transport distance is*

$$\inf_{T:T^{\#}P=Q} \int ||x - T(x)||^p dP(x) \tag{27}$$

*A minimizer $T^*$, if one exists, is called the optimal transport map.*

**Definition 4** (Wasserstein Distance, Earth Mover Distance, Optimal Transport Plan). *Let $\Gamma(\mu, \nu)$ denote all joint distributions $\gamma$ for $(X, Y)$ that have marginals $\mu$ and $\nu$. Then the Wasserstein distance is*

$$W_p(\mu, \nu) = \left( \inf_{\gamma \in \Gamma(\mu,\nu)} \int ||x - y||_2^p d\gamma(x, y) \right)^{\frac{1}{p}} \text{ where } p \geq 1 \tag{28}$$

*When $p = 1$, this is also called the Earth Mover distance. The minimizer $\gamma^*$ (which does exist) is called the optimal transport plan.*

**Lemma A.1** (Dual Formulation of Wasserstein Distance When $p \leq 1$). *It can be shown that*

$$W_p^p(\mu, \nu) = \sup_{\psi, \phi} \int \psi(y) d\nu(y) - \int \phi(x) d\mu(x) \tag{29}$$

*where $\psi(y) - \phi(x) \leq ||x - y||^p$. In the special case when $p = 1$, we have the very simple representation*

$$W_1(\mu, \nu) = \sup_{\varphi \in \mathcal{F}_1} \int \varphi(x) d\mu(x) - \int \varphi(x) d\nu(x) = \sup_{\varphi \in \mathcal{F}_1} \int \varphi(x) d(\mu - \nu)(x) = \sup_{\varphi \in \mathcal{F}_1} \int \varphi(x)(f - g)(x) dx \tag{30}$$

*where $\mathcal{F}_1$ denotes all maps from $\mathbb{R}^d$ to $\mathbb{R}$ such that $|f(x) - f(y)| \leq ||x - y||$ for all $x, y$. In the case when $0 < p < 1$, we have similar simple representation*

$$W_p(\mu, \nu) = \sup_{\varphi \in \mathcal{F}_p} \int \varphi(x) d\mu(x) - \int \varphi(x) d\nu(x) = \sup_{\varphi \in \mathcal{F}_p} \int \varphi(x) d(\mu - \nu)(x) = \sup_{\varphi \in \mathcal{F}_p} \int \varphi(x)(f - g)(x) dx \tag{31}$$

*where $\mathcal{F}_p$ denotes all maps from $\mathbb{R}^d$ to $\mathbb{R}$ such that $|f(x) - f(y)| \leq ||x - y||^p$ for all $x, y$.*

**Lemma A.2** (Dual Formulation of Wasserstein Distance When $1 < p < \infty$). *In the case when $1 < p < \infty$ and the support sets of measure $\mu$ and $\nu$ are included in a convex compact set $K$. Define $R = \sup_{x \in K} ||x||_2$, then we have slightly different dual formulation*

$$W_p(\mu, \nu) \geq \left( \sup_{\varphi \in Lip(p(2R)^{p-1})} \int \varphi(y) d(\nu - \mu)(y) - (p-1)(2R)^{p-1} \right)^{\frac{1}{p}}$$

$$= \left( \sup_{\varphi \in Lip(p(2R)^{p-1})} \int \varphi(y)(g - f)(y) dy - (p-1)(2R)^{p-1} \right)^{\frac{1}{p}} \tag{32}$$

*where $Lip(p(2R)^{p-1})$ denotes all maps $f$ from $\mathbb{R}^d$ to $\mathbb{R}$ such that $|f(x) - f(y)| \leq p(2R)^{p-1}||x - y||$ for all $x, y \in K$.*

**Definition 5** (Total Variation Distance). *The total variation distance between two probability distribution $\mu$ and $\nu$ on $\mathbb{R}^d$ is defined by*

$$||\mu - \nu||_{TV} = \max \left\{ |\mu(A) - \nu(A)| : A \subseteq \mathcal{R}^d \right\} \tag{33}$$

*where $\mathcal{R}^d$ is the set of all Borel subsets.*

**Lemma A.3.** *Let $\mu$ and $\nu$ be two probability distributions on $\mathbb{R}^d$ and absolutely continuous w.r.t. Lebesgue measure $\lambda$. Assume the density function of measure $\mu$ and $\nu$ w.r.t. $\lambda$ are $f(x)$ and $g(x)$. Then,*

$$||\mu - \nu||_{TV} = \frac{1}{2} \int_{\mathbb{R}^d} |f(x) - g(x)| dx \tag{34}$$

## B  PROOF OF LEMMA A.2

Recall the dual form of Wasserstein distance

$$W_p^p(\mu, \nu) = \sup_{\psi, \phi \in \mathcal{C}(\mathbb{R}^d)} \int \psi(y) d\nu(y) - \int \phi(x) d\mu(x) \tag{35}$$

where $\psi(y) - \phi(x) \leq ||x - y||^p$.

For simplicity of the proof, consider equivalent form

$$W_p^p(\mu, \nu) = \sup_{\psi, \phi \in \mathcal{C}(\mathbb{R}^d)} \int \psi(y) d\nu(y) + \int \phi(x) d\mu(x) \tag{36}$$

where $\psi(y) - \phi(x) \leq ||x - y||^p$. First, we introduce a theorem in Thorpe (2018)

**Theorem B.1** (Existence of a Maximiser to the Dual Problem). *Let $\mu \in \mathcal{P}(X), \nu \in \mathcal{P}(Y)$, where $X$ and $Y$ are polish, and $c : X \times Y \to [0, +\infty)$. Assume that there exists $c_X \in L^1(\mu), c_Y \in L^1(\nu)$ such that $c(x, y) \leq c_X(x) + c_Y(y)$ for $\mu$-almost every $x \in X$ and $\nu$-almost every $y \in Y$. In addition, assume that*

$$M := \int_X c_X(x) d\mu(x) + \int_Y c_Y(y) d\nu(y) < \infty \tag{37}$$

*Then there exists $(\varphi, \psi) \in \Phi_c = \{(\varphi, \psi) \in L^1(\mu) \times L^1(\nu) : \varphi(x) + \psi(y) \leq c(x, y)\}$ where the inequality is understood to hold for $\mu$-almost every $x \in X$ and $\nu$-almost every $y \in Y$ such that*

$$\sup_{\Phi_c} \mathbb{J} = \mathbb{J}(\varphi, \psi) \tag{38}$$

*where $\mathbb{J}$ is defined by $\mathbb{J} : L^1(\mu) \times L^1(\nu) \to \mathbb{R}, \mathbb{J}(\varphi, \psi) = \int_X \varphi d\mu + \int_Y \psi d\nu$. Futhermore we can choose $(\varphi, \psi) = (\eta^{cc}, \eta^c)$ for some $\eta \in L^1(\mu)$. For $\eta : X \to \mathbb{R}$, the c-transforms $\eta^c, \eta^{cc}$ are defined by*

$$\eta^c : Y \to \bar{\mathbb{R}}, \quad \eta^c(y) = \inf_{x \in X} (c(x, y) - \eta(x)) \tag{39}$$

$$\eta^{cc} : Y \to \bar{\mathbb{R}}, \quad \eta^{cc}(y) = \inf_{x \in X} (c(x, y) - \eta^c(x)) \tag{40}$$

**Lemma B.1.** *For $a, b \in \mathbb{R}$ and $1 \leq p < \infty$,*

$$|a + b|^p \leq 2^{p-1}(|a|^p + |b|^p) \tag{41}$$

*Proof.* First, it's easy to verify the cases when either of $a = 0, b = 0, p = 1$ holds. Then, Wlog, assume $a, b \in \mathbb{R}^+$

$$|a + b|^p \leq 2^{p-1}(|a|^p + |b|^p)$$
$$\iff (a + b)^p \leq 2^{p-1}(a^p + b^p)$$
$$\iff 2^{p-1}\left(\left(\frac{a}{a+b}\right)^p + \left(\frac{b}{a+b}\right)^p\right) \geq 1$$
$$\iff 2^{p-1}[x^p + (1-x)^p] \geq 1, \forall x \in (0, 1)$$

where the last inequality is easy to verify. $\qquad \square$

In our case, $c(x, y) = ||x - y||^p \leq (||x|| + ||y||)^p \leq 2^{p-1}(||x||^p + ||y||^p)$ and the requirement that $M < \infty$ is exactly the condition that $\mu$ and $\nu$ have finite $p^{\text{th}}$ moments which is easy to verify by noting that $\text{supp}(\mu) = \text{supp}(\nu) = K$ is compact set in $\mathbb{R}^d$. Then, according to the theorem, there exists $\eta \in L^1(\mu)$ such that

$$W_p^p(\mu, \nu) = \sup_{\eta \in L^1(\mu)} \int \eta^c(y) d\nu(y) + \int \eta^{cc}(x) d\mu(x) \tag{42}$$

Note that $\eta^c$ possesses Lipschitz continuous property stated below

**Lemma B.2.** *For $\eta \in L^1(K)$ where $K \subseteq \mathbb{R}^d$ is a convex compact set, then $\eta^{c_p}$ is a $p(2R)^{p-1}$-Lipschitz function where $R := \sup_{x \in K} ||x||$ and $c_p(x, y) = ||x - y||^p$, i.e.,*

$$||\eta^{c_p}(x) - \eta^{c_p}(y)|| \leq p(2R)^{p-1}||x - y||, \quad x, y \in K \tag{43}$$

*Proof.*

$$|\eta^{c_p}(x) - \eta^{c_p}(y)| = \left| \inf_{z_1 \in K} \left( ||x - z_1||^p - \eta(z_1) \right) - \inf_{z_2 \in K} \left( ||y - z_2||^p - \eta(z_2) \right) \right| \tag{44}$$

$$= \left| \inf_{z_1 \in K} \sup_{z_2 \in K} \left( \left( ||x - z_1||^p - ||y - z_2||^p \right) - \left( \eta(z_1) - \eta(z_2) \right) \right) \right|$$

$$\leq \sup_{z \in K} \left| \left( ||x - z||^p - \eta(z) \right) - \left( ||y - z||^p - \eta(z) \right) \right| \tag{45}$$

$$= \sup_{z \in K} \left| ||x - z||^p - ||y - z||^p \right|$$

where 44 is due to the definition of c-transform; 45 is obtained by taking a specific value of $z_1$ as $z_2$. Note that $K$ is a compact set and $\left| ||x - z||^p - ||y - z||^p \right|$ is a continuous function w.r.t. $z$, then there exists a point $z^*$ such that $\left| ||x - z^*||^p - ||y - z^*||^p \right| = \sup_{z \in K} \left| ||x - z||^p - ||y - z||^p \right|$. According to the first order condition, $z^*$ satisfies the equation below

$$\nabla_z(||x - z||^p - ||y - z||^p) = \nabla_z||x - z||^p - \nabla_z||y - z||^p = \nabla_z||z - x||^p - \nabla_z||z - x||^p$$

$$= p||z - x||^{\frac{p}{2}-1}(z - x)^\top - p||z - y||^{\frac{p}{2}-1}(z - y)^\top = 0 \tag{46}$$

$$\Longrightarrow ||z - x||^{\frac{p}{2}-1}(z - x)^\top = ||z - y||^{\frac{p}{2}-1}(z - y)^\top$$

$$\Longrightarrow (||z - x||^{\frac{p}{2}-1} - ||z - y||^{\frac{p}{2}-1})z^\top = ||z - x||^{\frac{p}{2}-1}x^\top - ||z - y||^{\frac{p}{2}-1}y^\top$$

$$\Longrightarrow z = \frac{||z - x||^{\frac{p}{2}-1}}{||z - x||^{\frac{p}{2}-1} - ||z - y||^{\frac{p}{2}-1}}x - \frac{||z - y||^{\frac{p}{2}-1}}{||z - x||^{\frac{p}{2}-1} - ||z - y||^{\frac{p}{2}-1}}y$$

where 46 is due to $\nabla_x||x||^p = \nabla_x(x^\top x)^{\frac{p}{2}} = p(x^\top x)^{\frac{p}{2}-1}x^\top = p||x||^{\frac{p}{2}-1}x^\top$. And this equation shows that $z^*$ lie on the line determined by $x$ and $y$ but does not lies on the part between $x$ and $y$, which can be formulated as $z^* = \lambda x + (1 - \lambda)y, \lambda \in \mathbb{R} \setminus (0, 1)$. Note that

$$\sup_{\lambda \in \mathbb{R} \setminus (0,1)} \left| ||x - \left( \lambda x + (1 - \lambda)y \right)||^p - ||y - \left( \lambda x + (1 - \lambda)y \right)||^p \right|$$

$$= \sup_{\lambda \in \mathbb{R} \setminus (0,1)} \left| ||(1 - \lambda)(x - y)||^p - ||\lambda(y - x)||^p \right|$$

$$= \sup_{\lambda \in \mathbb{R} \setminus (0,1)} \left| |1 - \lambda|^p - |\lambda|^p \right| \cdot ||x - y||^p = \left( \sup_{\lambda \in \mathbb{R} \setminus (0,1)} \left| |1 - \lambda|^p - |\lambda|^p \right| \right) \cdot ||x - y||^p$$

Then, we just need to optimize

$$\sup_{\lambda \in \mathbb{R} \setminus (0,1)} \left| |1 - \lambda|^p - |\lambda|^p \right|$$

$$s.t. \quad \lambda x + (1 - \lambda)y \in K$$

Note that we can relax the constraint as below

$$\lambda x + (1 - \lambda)y \in K$$

$$\iff \lambda(x - y) + y = (1 - \lambda)(y - x) + x \in K$$

$$\Longrightarrow ||\lambda(x - y) + y|| = ||(1 - \lambda)(y - x) + x|| \leq R \tag{47}$$

$$\Longrightarrow ||\lambda(x - y)|| \leq R + ||y||, \quad ||(1 - \lambda)(y - x)|| \leq R + ||x|| \tag{48}$$

$$\Longrightarrow |\lambda| \cdot ||x - y|| \leq 2R, \quad |1 - \lambda| \cdot ||x - y|| \leq 2R \tag{49}$$

$$\Longrightarrow 1 - \frac{2R}{||x - y||} \leq \lambda \leq \frac{2R}{||x - y||}$$

where 47 and 49 is due to the definition of $R$ as $\sup_{x \in K} ||x||$; 48 is due to triangular inequality.

Using the relaxed constraint, we can show that when $\lambda \geq 1$, $\left| |1 - \lambda|^p - |\lambda|^p \right| = \lambda^p - (\lambda - 1)^p$ is an increasing function w.r.t. $\lambda$ as $p \geq 1$, then

$$\left| |1 - \lambda|^p - |\lambda|^p \right| = \lambda^p - (\lambda - 1)^p \leq \left( \frac{2R}{||x - y||} \right)^p - \left( \frac{2R}{||x - y||} - 1 \right)^p \tag{50}$$

And when $\lambda \leq 0$, $\left| |1 - \lambda|^p - |\lambda|^p \right| = (1 - \lambda)^p - (-\lambda)^p$ is a decreasing function w.r.t $\lambda$ as $p \geq 1$, then

$$\left| |1 - \lambda|^p - |\lambda|^p \right| = (1 - \lambda)^p - (-\lambda)^p \leq \left( \frac{2R}{||x - y||} \right)^p - \left( \frac{2R}{||x - y||} - 1 \right)^p \tag{51}$$

Note that

$$\left( \frac{2R}{||x - y||} \right)^p - \left( \frac{2R}{||x - y||} - 1 \right)^p = p\left( k \cdot \left( \frac{2R}{||x - y||} \right) + (1 - k) \cdot \left( \frac{2R}{||x - y||} - 1 \right) \right)^{p-1} \tag{52}$$

$$= p\left( \frac{2R}{||x - y||} + (k - 1) \right)^{p-1} \leq p\left( \frac{2R}{||x - y||} \right)^{p-1}$$

where 52 is due to the Differential Mean Value Theorem where $k \in (0, 1)$.

Thus, we have

$$|\eta^{c_p}(x) - \eta^{c_p}(y)| \leq p\left( \frac{2R}{||x - y||} \right)^{p-1} \cdot ||x - y||^p = p(2R)^{p-1}||x - y|| \tag{53}$$

i.e. $\eta^{c_p}(x)$ is a $p(2R)^{p-1}$-Lipschitz function. $\qquad\qquad\square$

Using Lipschitz continuous property of $\eta^c$, we get

$$W_p^p(\mu, \nu) = \sup_{\eta \in L^1(\mu)} \int \eta^c(y)d\nu(y) + \int \eta^{cc}(x)d\mu(x) \leq \sup_{\varphi \in \mathbf{Lip}(p(2R)^{p-1})} \int \varphi(y)d\nu(y) + \int \varphi^c(x)d\mu(x)$$

$$= \sup_{\varphi \in \mathbf{Lip}(p(2R)^{p-1})} \int \varphi(y)d\nu(y) + \int \varphi^c(y)d\mu(y) \tag{54}$$

where $\mathbf{Lip}(p(2R)^{p-1})$ denotes the set of $p(2R)^{p-1}$-Lipschitz functions. On the other hand, recall that

$$W_p^p(\mu, \nu) = \sup_{\psi, \phi \in \mathcal{C}(\mathbb{R}^d)} \int \psi(y)d\nu(y) + \int \phi(x)d\mu(x) \tag{55}$$

where $\psi(y) + \phi(x) \leq ||x - y||^p$. Keeping $\psi(x)$ fixed and optimizing w.r.t. $\phi(y)$, then we just need to optimize $\int \phi(y)d\mu(y)$ under constraint $\phi(y) \leq ||x - y||^p - \psi(x)$. Then obviously we have $\phi^*(y) = \inf_{x \in K} \left( ||x - y||^p - \psi(x) \right) = \psi^{c_p}(y)$ where $c_p(x, y) = ||x - y||^p$. The map $(\phi, \psi) \in \mathcal{C}(K)^2 \mapsto (\psi^{c_p}, \psi) \in \mathcal{C}(K)^2$ replaces dual potentials by "better" ones improving the dual objective $W_p^p(\mu, \nu)$.

Using $c$-transform, we can reformulate constrained problem into unconstrained convex problem over a single potential

$$W_p^p(\mu, \nu) = \sup_{\psi \in \mathcal{C}(\mathbb{R}^d)} \int \psi(y)d\nu(y) + \int \psi^{c_p}(x)d\mu(x) = \sup_{\psi \in \mathcal{C}(\mathbb{R}^d)} \int \psi(y)d\nu(y) + \int \psi^{c_p}(y)d\mu(y) \tag{56}$$

Combining 54 and 56, we know that when the support set of measure $\mu$ and $\nu$ $\mathrm{supp}(\mu) = \mathrm{supp}(\nu) = K$ where $K$ is a convex compact set, we have

$$\sup_{\psi \in \mathcal{C}(K)} \int \psi(y)d\nu(y) + \int \psi^{c_p}(y)d\mu(y) = W_p^p(\mu, \nu) \leq \sup_{\varphi \in \mathbf{Lip}(p(2R)^{p-1})} \int \varphi(y)d\nu(y) + \int \varphi^c(y)d\mu(y) \tag{57}$$

Note that Lipschitz function must be continuous and therefore $\mathbf{Lip}(p(2R)^{p-1}) \subseteq \mathcal{C}(K)$. Then, we have

$$\sup_{\varphi \in \mathbf{Lip}(p(2R)^{p-1})} \int \varphi(y)d\nu(y) + \int \varphi^c(y)d\mu(y) \leq \sup_{\phi \in \mathcal{C}(K)} \int \psi(y)d\nu(y) + \int \psi^{c_p}(y)d\mu(y) \tag{58}$$

Combining 57 and 58, we know the inequality in 57 changes into equality

$$W_p^p(\mu, \nu) = \sup_{\varphi \in \mathbf{Lip}(p(2R)^{p-1})} \int \varphi(y) d\nu(y) + \int \varphi^c(y) d\mu(y) \tag{59}$$

Note that for $f(x) = x^p - p(2R)^{p-1}x, x \in \mathbb{R}^+$ achieves its minimum when $f'(x) = px^{p-1} - p(2R)^{p-1} = 0$, i.e. $x = 2R$ and the minimum is $f(2R) = -(p-1)(2R)^{p-1}$. Then,

$$\varphi^{c_p}(y) = \inf_{x \in K} \left( ||x-y||^p - \varphi(x) \right) \geq \inf_{x \in K} \left( ||x-y||^p - \varphi(y) - p(2R)^{p-1}||x-y|| \right) = -\varphi(y) - (p-1)(2R)^{p-1} \tag{60}$$

Thus, we attain a lower bound of $W_p^p(\mu, \nu)$

$$W_p^p(\mu, \nu) = \sup_{\varphi \in \mathbf{Lip}(p(2R)^{p-1})} \int \varphi(y) d\nu(y) + \int \varphi^c(y) d\mu(y) \tag{61}$$

$$\geq \sup_{\varphi \in \mathbf{Lip}(p(2R)^{p-1})} \int \varphi(y) d\nu(y) - \int \left( \varphi(y) + (p-1)(2R)^{p-1} \right) d\mu(y) \tag{62}$$

$$= \sup_{\varphi \in \mathbf{Lip}(p(2R)^{p-1})} \int \varphi(y) d\nu(y) - \int \varphi(y) d\mu(y) - (p-1)(2R)^{p-1}$$

$$= \sup_{\varphi \in \mathbf{Lip}(p(2R)^{p-1})} \int \varphi(y) d(\nu - \mu)(y) - (p-1)(2R)^{p-1}$$

where 61 is due to 59 and 62 is due to 60.

## C  PROOF OF THEOREM 3.1

*Proof.*
$$\mathcal{D}_{x,\epsilon,q} = \{x' + \mu : ||x - x'||_q \leq \epsilon\} \tag{63}$$

Note that

$$\sup_{\nu \in \mathcal{D}_{x,\epsilon,q}} W_p(\mu, \nu) = \sup_{||x-x'||_q \leq \epsilon} W_p(x + \mu, x' + \mu) = \sup_{||z||_q \leq \epsilon} W_p(\mu, z + \mu) \tag{64}$$

where the first equality is due to the definition of $\mathcal{D}_{x,\epsilon,q}$ and the second equality is due to the translation invariance property of Wasserstein distance.

Then recall the Monge version of Wasserstein distance

$$W_p(\mu, \nu) \leq \left( \inf_{T:T^\# \mu = \nu} \int ||x - T(x)||^p d\mu(x) \right)^{\frac{1}{p}} \tag{65}$$

Noticing the **inf** operator in the Monge version definition of $W_p$, we can get an upper bound for $W_p(\mu, \nu)$ by specializing a transport map $\tilde{T}$ satisfying $\tilde{T}\mu = \nu$. In our case, we take $\tilde{T} : \mathbb{R}^d \to \mathbb{R}^d, \tilde{T} : x \mapsto x + z$, and it's easy to verify that $\tilde{T}^\# \mu = z + \mu$. Then we get the upper bound below

$$W_p(\mu, z + \mu) \leq \left( \inf_{T:T^\# \mu = z + \mu} \int ||x - T(x)||^p d\mu(x) \right)^{\frac{1}{p}} \leq \left( \int ||x - \tilde{T}(x)||^p d\mu(x) \right)^{\frac{1}{p}} = ||z|| \tag{66}$$

where the last equality is due to $\mu$ is a probability measure. This provides us with an intuition that the upper bound of $W_p(\mu, z + \mu)$ is determined by the Euclidean norm of displacement $z$. Using this upper bound,

$$\sup_{||z||_q \leq \epsilon} W_p(\mu, z + \mu) \leq \sup_{||z||_q \leq \epsilon} ||z||_2 \tag{67}$$

When $0 < q \leq 2$, using the lemma that when $0 < p < q < \infty$, $||z||_q \leq ||z||_p, \forall z \in \mathbb{R}^d$ holds, we have $\sup_{||z||_q \leq \epsilon} ||z||_2 \leq \sup_{||z||_q \leq \epsilon} ||z||_q = \epsilon$. On the other hand, note that $||\epsilon e_1||_2 = ||\epsilon e_1||_q = \epsilon$, we have $\sup_{||z||_q \leq \epsilon} ||z||_2 = \epsilon$. And when $q > 2$, recall Holder's Inequality below

**Lemma C.1** (Holder's Inequality for $R^n$)**.** *For $\{a_i\}_{1 \leq i \leq n}, \{b_i\}_{1 \leq i \leq n} \subseteq \mathbb{R}, r > 1$, we have*

$$\sum_{i=1}^{n} |a_i||b_i| \leq \left( \sum_{i=1}^{n} |a_i|^r \right)^{\frac{1}{r}} \left( \sum_{i=1}^{n} |a_i|^{\frac{r}{r-1}} \right)^{\frac{r-1}{r}} \tag{68}$$

Apply it to the case $n = d, |a_i| = |x_i|^2, |b_i| = 1$ and $r = \frac{q}{2} > 1$,

$$\sum_{i=1}^{d} |x_i|^2 = \sum_{i=1}^{d} |x_i|^2 \cdot 1 \leq \left( \sum_{i=1}^{d} (|x_i|^2)^{\frac{q}{2}} \right)^{\frac{2}{q}} \left( \sum_{i=1}^{d} 1^{\frac{q}{q-2}} \right)^{\frac{q-2}{q}} = \left( \sum_{i=1}^{d} |x_i|^q \right)^{\frac{2}{q}} d^{1-\frac{2}{q}} \quad (69)$$

$$||x||_2 = \left( \sum_{i=1}^{d} |x_i|^2 \right)^{\frac{1}{2}} \leq \left( \sum_{i=1}^{d} |x_i|^q \right)^{\frac{1}{q}} d^{\frac{1}{2}-\frac{1}{q}} = ||x||_q d^{\frac{1}{2}-\frac{1}{q}} \quad (70)$$

Thus, $\sup_{||z||_q \leq \epsilon} ||z||_2 \leq \sup_{||z||_q \leq \epsilon} ||x||_q d^{\frac{1}{2}-\frac{1}{q}} = \epsilon d^{\frac{1}{2}-\frac{1}{q}}$. On the other hand, note that $||\frac{\epsilon}{n^{\frac{1}{q}}} \sum_{i=1}^{d} e_i||_q = \epsilon, ||\frac{\epsilon}{n^{\frac{1}{q}}} \sum_{i=1}^{d} e_i||_2 = \epsilon d^{\frac{1}{2}-\frac{1}{q}}$, we have $\sup_{||z||_q \leq \epsilon} ||z||_2 = \epsilon d^{\frac{1}{2}-\frac{1}{q}}$. Combining the case when $0 < q \leq 2$ and $q > 2$, we have

$$\sup_{||x-x'||_q \leq \epsilon} W_p(x + \mu, x' + \mu) = \sup_{||z||_q \leq \epsilon} W_p(\mu, z + \mu) \leq \begin{cases} \epsilon \text{ when } 0 < q \leq 2 \\ \epsilon d^{\frac{1}{2}-\frac{1}{q}} \text{ when } q > 2 \end{cases} = \max\{\epsilon, \epsilon d^{\frac{1}{2}-\frac{1}{q}}\} \quad (71)$$

$\square$

# D $W_2$ DISTANCE RELAXATION IS TIGHT FOR GAUSSIAN PROBABILITY MEASURE

Here, we show that $W_2$ distance relaxation for Gaussian probability measure is tight.

**Theorem D.1.** *When $\mu = \mathcal{N}(0, \sigma^2 I)$ and $p = 2$, the relaxation in 9 is tight. In other words,*

$$\mathcal{D}_{x,\epsilon,q} \subseteq \mathcal{D}_{x,\max\{\epsilon, \epsilon d^{\frac{1}{2}-\frac{1}{q}}\},2} \text{ but } \mathcal{D}_{x,\epsilon,q} \setminus \mathcal{D}_{x,\max\{\epsilon, \epsilon d^{\frac{1}{2}-\frac{1}{q}}\}-\delta,2} \neq \emptyset \text{ for any sufficiently small } \delta > 0. \quad (72)$$

*Proof.* Note that Dowson & Landau (1982) established the formula of Wasserstein distance between two Gaussian measures.

**Theorem D.2.** *For Gaussian probability measures $\mu = \mathcal{N}(\mu_1, \Sigma_1)$ and $\nu = \mathcal{N}(\mu_2, \Sigma_2)$, $W_2$-distance between $\mu$ and $\nu$ have closed form formula*

$$W_2(\mu, \nu)^2 = ||\mu_1 - \mu_2||^2 + tr\left(\Sigma_1 + \Sigma_2 - 2(\Sigma_1 \Sigma_2)^{\frac{1}{2}}\right) \quad (73)$$

Using above theorem, we yield following tight relaxation between norm-based constraint set $\mathcal{D}_{x,\epsilon,q}$ and $W_2$-distance based constraint sets $\mathcal{D}_{x,\delta,2}$ for Gaussian smoothing measures centered at origin, i.e. $\mu = \mathcal{N}(0, \sigma^2 I)$

$$\sup_{\nu \in \mathcal{D}_{x,\epsilon,q}} W_2(\mu, \nu) = \sup_{||x-x'||_q \leq \epsilon} W_2(x + \mu, x' + \mu) = \sup_{||z||_q \leq \epsilon} W_2(\mu, z + \mu)$$

$$= \sup_{||z||_q \leq \epsilon} ||z||_2 = \max\{\epsilon, \epsilon d^{\frac{1}{2}-\frac{1}{q}}\} \quad (74)$$

where 74 is due to theorem D.2 and equality 71. And generalization of above theorem when $\mu = \mathcal{N}(0, \Sigma)$ can be proved in the same way. $\square$

# E PROOF OF THEOREM 3.2

*Proof.* First, we introduce the lemma below.

**Lemma E.1.** *Let $X$ be a random variable that follows $d$-dimensional Gaussian distribution with density function*

$$f(x; \mu, \Sigma) = \frac{1}{(2\pi)^{\frac{d}{2}} |\Sigma|^{\frac{1}{2}}} e^{-\frac{1}{2}(x-\mu)^T \Sigma^{-1}(x-\mu)} \quad (75)$$

*where $x, \mu \in \mathbb{R}^d$ and $\Sigma \in S_{++}^d$. Let $H : x^T w + b = 0$ be a hyperplane in the $d$-dimensional Euclidean space $\mathbb{R}^d$, where $w \in \mathbb{R}^d$ and $b \in \mathbb{R}$. The hyperplane $H$ defines two half-spaces:*

$$\Omega_+ = \{x \in \mathbb{R}^d | x^T w + b \geq 0\}, \quad \Omega_- = \{x \in \mathbb{R}^d | x^T w + b < 0\} \quad (76)$$

*Define the integral over half-space $\Omega_+$ as*

$$P = \int_{\Omega_+} f(x; \mu, \Sigma) dx = \int_{\Omega_+} \frac{1}{(2\pi)^{\frac{d}{2}}|\Sigma|^{\frac{1}{2}}} e^{-\frac{1}{2}(x-\mu)^T \Sigma^{-1}(x-\mu)} dx$$

*Since $\Sigma$ is positive definite symmetric, there exist an orthogonal matrix $U$ and a diagonal matrix $D$ with positive diagonal elements such that $\Sigma = U^T DU$. Let $x_0 = -\frac{\mu^\top w + b}{||\sqrt{DU}w||_2}$ and hence $P = \int_{x_0}^{\infty} \frac{1}{\sqrt{2\pi}} e^{-\frac{1}{2}x^2} dx$.*

(The proof of this lemma is credit to `https://math.stackexchange.com/questions/556977/gaussian-integrals-over-a-half-space`.)

Recall the definition of $l_p$-norm constraint set of probability measures

$$\mathcal{D}_{x,\epsilon,q} = \{x' + \mu : ||x - x'||_q \le \epsilon\} \tag{77}$$

Note that

$$\sup_{\nu \in \mathbb{D}_{x,\epsilon,q}} TV(\mu, \nu) = \sup_{||x-x'||_q \le \epsilon} TV(x + \mu, x' + \mu) = \sup_{||z||_q \le \epsilon} TV(\mu, z + \mu) \tag{78}$$

where the first equality is due to the definition of $\mathcal{D}_{x,\epsilon,q}$ and the second equality is due to the translation invariance property of total variance distance.

Define hyperplane $H^1 : x^T z - \frac{||z||_2^2}{2} = 0$ and $H^2 : x^T z + \frac{||z||_2^2}{2} = 0$. The hyperplane $H^1$ defines two half-spaces: $\Omega_+^1 = \{x \in \mathbb{R}^d | x^T z - \frac{||z||_2^2}{2} \ge 0\}$ and $\Omega_-^1 = \{x \in \mathbb{R}^d | x^T z - \frac{||z||_2^2}{2} < 0\}$. And the hyperplane $H^2$ defines two half-spaces: $\Omega_+^2 = \{x \in \mathbb{R}^d | x^T z + \frac{||z||_2^2}{2} \ge 0\}$ and $\Omega_-^2 = \{x \in \mathbb{R}^d | x^T z + \frac{||z||_2^2}{2} < 0\}$. Applying lemma E.1 and lemma A.3, we know that

$$\sup_{||z||_q \le \epsilon} TV(\mu, z + \mu)$$

$$= \sup_{||z||_q \le \epsilon} \frac{1}{2} \int \left| \frac{1}{(2\pi)^{\frac{d}{2}}\sigma^d} e^{-\frac{x^T x}{2\sigma^2}} - \frac{1}{(2\pi)^{\frac{d}{2}}\sigma^d} e^{-\frac{(x-z)^T(x-z)}{2\sigma^2}} \right| dx \tag{79}$$

$$= \frac{1}{2} \sup_{||z||_q \le \epsilon} \int_{\Omega_+^1} \frac{1}{(2\pi)^{\frac{d}{2}}\sigma^d} \left| e^{-\frac{x^T x}{2\sigma^2}} - e^{-\frac{(x-z)^T(x-z)}{2\sigma^2}} \right| dx + \int_{\Omega_-^1} \frac{1}{(2\pi)^{\frac{d}{2}}\sigma^d} \left| e^{-\frac{x^T x}{2\sigma^2}} - e^{-\frac{(x-z)^T(x-z)}{2\sigma^2}} \right| dx$$

$$= \frac{1}{2} \sup_{||z||_q \le \epsilon} \int_{\Omega_+^1} \frac{1}{(2\pi)^{\frac{d}{2}}\sigma^d} \left( e^{-\frac{(x-z)^T(x-z)}{2\sigma^2}} - e^{-\frac{x^T x}{2\sigma^2}} \right) dx + \int_{\Omega_-^1} \frac{1}{(2\pi)^{\frac{d}{2}}\sigma^d} \left( e^{-\frac{x^T x}{2\sigma^2}} - e^{-\frac{(x-z)^T(x-z)}{2\sigma^2}} \right) dx \tag{80}$$

$$= \frac{1}{2} \sup_{||z||_q \le \epsilon} \int_{\Omega_+^1} d(z + \mu) - \int_{\Omega_+^1} d\mu + \int_{\Omega_-^1} d\mu - \int_{\Omega_-^1} d(z + \mu)$$

$$= \frac{1}{2} \sup_{||z||_q \le \epsilon} \int_{\Omega_+^2} d\mu - \int_{\Omega_+^1} d\mu + \int_{\Omega_-^1} d\mu - \int_{\Omega_-^2} d\mu \tag{81}$$

$$= \frac{1}{2} \sup_{||z||_q \le \epsilon} \int_{-\frac{||z||_2}{2\sigma}}^{\infty} \frac{1}{\sqrt{2\pi}} e^{-\frac{1}{2}x^2} dx - \int_{\frac{||z||_2}{2\sigma}}^{\infty} \frac{1}{\sqrt{2\pi}} e^{-\frac{1}{2}x^2} dx + \int_{-\infty}^{\frac{||z||_2}{2\sigma}} \frac{1}{\sqrt{2\pi}} e^{-\frac{1}{2}x^2} dx - \int_{-\infty}^{-\frac{||z||_2}{2\sigma}} \frac{1}{\sqrt{2\pi}} e^{-\frac{1}{2}x^2} dx \tag{82}$$

$$= \frac{1}{2} \sup_{||z||_q \le \epsilon} \left( G\left(\frac{||z||_2}{2\sigma}\right) - G\left(-\frac{||z||_2}{2\sigma}\right) + G\left(\frac{||z||_2}{2\sigma}\right) - G\left(-\frac{||z||_2}{2\sigma}\right) \right) \tag{83}$$

$$= \frac{1}{2} \sup_{||z||_q \le \epsilon} 2\left( 2G\left(\frac{||z||_2}{2\sigma}\right) - 1 \right) \tag{84}$$

$$= 2G\left(\frac{\max\{\epsilon, \epsilon d^{\frac{1}{2} - \frac{1}{q}}\}}{2\sigma}\right) - 1 \tag{85}$$

where 79 is due to lemma A.3; 80 is due to the consistency of sign of integrand function on $\Omega_+^1$ and $\Omega_-^1$; 81 is due to the transformation formula of space coordinates; 82 is due to lemma E.1; 83 and 85 is due to the definition and central symmetry property of $G$ as the cumulative density function of standard normal distribution; 85 is due to 71. $\qquad \square$

## F    PROOF OF THEOREM 3.3

*Proof.* Recall the definition of $l_p$-norm constraint set of probability measures

$$\mathcal{D}_{x,\epsilon,q} = \{x' + \mu : ||x - x'||_q \le \epsilon\} \tag{86}$$

Note that

$$\sup_{\nu \in \mathcal{D}_{x,\epsilon,q}} TV(\mu, \nu) = \sup_{||x-x'||_q \le \epsilon} TV(x + \mu, x' + \mu) = \sup_{||z||_q \le \epsilon} TV(\mu, z + \mu) \tag{87}$$

where the first equality is due to the definition of $\mathcal{D}_{x,\epsilon,q}$ and the second equality is due to the translation invariance property of total variance distance. Next, compute the value of $TV(\mu, z + \mu)$.

**Lemma F.1.** *$K$ is a $l_1$ norm ball centered at original point of radius $r$, then $K \cap (z + K) = \emptyset$ if and only if $||z||_1 > 2r$.*

*Proof.* First, we prove the if part and assume $||z||_1 > 2r$. Consider arbitrarily taken $x \in (z + K)$, i.e. $||x - z||_1 \le r$. According to the triangular inequality with respect to $l_1$ norm, we have

$$||x||_1 = ||z - (x - z)||_1 \ge ||z||_1 - ||x - z||_1 > 2r - r = r \tag{88}$$

which shows that $x \notin K$ and therefore $K \cap (z + K) = \emptyset$.
Then we prove the only if part by using reduction to absurdity and assume $||z||_1 \le 2r$. Take $y = \frac{1}{2}z$, then $||y||_1 = \frac{1}{2}||z||_1 \le r$ and $||y - z||_1 = \frac{1}{2}||z||_1 \le r$ which shows that $y \in K \cap (z + K)$ and therefore $K \cap (z + K) \ne \emptyset$ which leads to a contradiction. $\square$

According to lemma F.1, we know that when $\{z | ||z||_q \le \epsilon, ||z||_1 \ge 2r\} \ne \emptyset$, we have

$$\sup_{\nu \in \mathcal{D}_{x,\epsilon,q}} TV(\mu, \nu) = \sup_{||z||_q \le \epsilon} TV(\mu, z + \mu) = 1 \tag{89}$$

Define $\bar{z} = \frac{2r}{d} \sum_{i=1}^d e_i$, and it's easy to verify that $||\bar{z}||_1 = 2r$ and $||\bar{z}||_q = 2rd^{\frac{1}{q}-1}$ for $q > 1$. Thus, when $\epsilon > 2rd^{\frac{1}{q}-1}$, we have

$$\sup_{\nu \in \mathcal{D}_{x,\epsilon,q}} TV(\mu, \nu) = \sup_{||z||_q \le \epsilon} TV(\mu, z + \mu) = 1 \tag{90}$$

$\square$

## G    PROOF OF THEOREM 3.4

*Proof.* First, we introduce the lemmas below for the convenience of later proof.

**Lemma G.1** (Volume Formula of $d$-dimensional spherical cap)**.** *The volume of a $d$-dimensional hyperspherical cap of height $h$ and radius $r$ is given by:*

$$V = \frac{\pi^{\frac{d-1}{2}} r^d}{\Gamma(\frac{d+1}{2})} \int_0^{arccos(\frac{r-h}{r})} \sin^d(t)dt \tag{91}$$

*where we define $h$ as the value shown in figure 4 and $\Gamma$ (the gamma function) is given by $\Gamma(z) = \int_0^\infty t^{z-1}e^{-t}dt$.*

**Lemma G.2** (Volume formula of $d$-dimensional Euclidean ball)**.** *The volume of $d$-dimensional Euclidean ball of radius $r$ is given by*

$$V = \frac{\pi^{\frac{d}{2}} r^d}{\Gamma(\frac{d}{2} + 1)} \tag{92}$$

Recall the definition of $l_p$-norm constraint set of probability measures

$$\mathcal{D}_{x,\epsilon,q} = \{x' + \mu : ||x - x'||_q \le \epsilon\} \tag{93}$$

Note that

$$\sup_{\nu \in \mathcal{D}_{x,\epsilon,q}} TV(\mu, \nu) = \sup_{||x-x'||_q \le \epsilon} TV(x + \mu, x' + \mu) = \sup_{||z||_q \le \epsilon} TV(\mu, z + \mu) \tag{94}$$

where the first equality is due to the definition of $\mathcal{D}_{x,\epsilon,q}$ and the second equality is due to the translation invariance property of total variance distance.

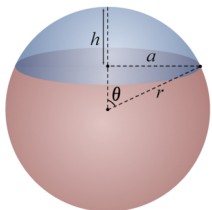

Figure 4: An example of a spherical cap in blue

**Lemma G.3.** *$K$ is a $l_2$ norm ball centered at original point of radius $r$, then $K \cap (z + K) = \emptyset$ if and only if $||z||_2 > 2r$.*

According to this lemma, we know that when $q \le 2$ and $\epsilon > 2r$, we have

$$1 \ge \sup_{||z||_q \le \epsilon} TV(\mu, z + \mu) \ge TV(\mu, \epsilon e_1 + \mu) = \frac{\text{Vol}(K\Delta(\epsilon e_1 + K))}{2\text{Vol}(K)} = 1 \tag{95}$$

where the last equality is due to $||\epsilon e_1||_2 = \epsilon > 2r$ and applying lemma G.3. And when $q > 2$ and $\epsilon > 2r d^{\frac{1}{q} - \frac{1}{2}}$, we have

$$1 \ge \sup_{||z||_q \le \epsilon} TV(\mu, z + \mu) \ge TV(\mu, \frac{\epsilon}{d^{\frac{1}{q}}} \sum_{i=1}^{d} e_i + \mu) = \frac{\text{Vol}(K\Delta(\frac{\epsilon}{d^{\frac{1}{q}}} \sum_{i=1}^{d} e_i + K))}{2\text{Vol}(K)} = 1 \tag{96}$$

where the last equality is due to $||\frac{\epsilon}{d^{\frac{1}{q}}} \sum_{i=1}^{d} e_i||_2 = \epsilon d^{\frac{1}{2} - \frac{1}{q}} > 2r$ and applying lemma G.3. Combining the results for $q \le 2$ and $q > 2$, we have

$$\sup_{||z||_q \le \epsilon} TV(\mu, z + \mu) = 1 \text{ when } \epsilon > \min\{2r, 2r d^{\frac{1}{q} - \frac{1}{2}}\} \tag{97}$$

Next, consider the case when $\epsilon \le \min\{2r, 2r d^{\frac{1}{q} - \frac{1}{2}}\}$. Applying G.1, lemma G.2 and lemma A.3, we have

$$\sup_{||z||_q \le \epsilon} TV(\mu, z + \mu)$$

$$= \sup_{||z||_q \le \epsilon} \frac{1}{2} \int \left| \frac{1}{\text{Vol}(K)} \mathcal{I}_{x \in K} - \frac{1}{\text{Vol}(K)} \mathcal{I}_{x \in z + K} \right| dx \tag{98}$$

$$= \sup_{||z||_q \le \epsilon} \frac{1}{2\text{Vol}(K)} \int \mathcal{I}_{x \in K\Delta(z+K)} dx = \sup_{||z||_q \le \epsilon} \frac{\text{Vol}(K\Delta(z+K))}{2\text{Vol}(K)}$$

$$= \sup_{||z||_q \le \epsilon} \frac{\text{Vol}(K) - \frac{2\pi^{\frac{d-1}{2}} r^d}{\Gamma(\frac{d+1}{2})} \int_0^{\arccos(\frac{||z||_2}{2r})} \sin^d(t) dt}{\text{Vol}(K)} = \sup_{||z||_q \le \epsilon} 1 - \frac{\frac{2\pi^{\frac{d-1}{2}} r^d}{\Gamma(\frac{d+1}{2})} \int_0^{\arccos(\frac{||z||_2}{2r})} \sin^d(t) dt}{\text{Vol}(K)} \tag{99}$$

$$= \sup_{||z||_q \le \epsilon} 1 - \frac{\frac{2\pi^{\frac{d-1}{2}} r^d}{\Gamma(\frac{d+1}{2})} \int_0^{\arccos(\frac{||z||_2}{2r})} \sin^d(t) dt}{\frac{\pi^{\frac{d}{2}}}{\Gamma(\frac{d}{2}+1)} r^d} = \sup_{||z||_q \le \epsilon} 1 - \frac{2\Gamma(\frac{d}{2} + 1)}{\pi^{\frac{1}{2}} \Gamma(\frac{d+1}{2})} \int_0^{\arccos(\frac{||z||_2}{2r})} \sin^d(t) dt \tag{100}$$

$$= 1 - \frac{2\Gamma(\frac{d}{2} + 1)}{\pi^{\frac{1}{2}} \Gamma(\frac{d+1}{2})} \int_0^{\arccos(\frac{\max\{\epsilon, \epsilon d^{\frac{1}{2} - \frac{1}{q}}\}}{2r})} \sin^d(t) dt \tag{101}$$

where 114 is due to lemma A.3; 99 is due to lemma G.1; 100 is due to lemma G.2; 101 is due to 71. Because of the computation difficulty (overflow), we have to simplify the term $\frac{\Gamma(\frac{d}{2}+1)}{\Gamma(\frac{d+1}{2})}$.

When $d$ is even, assume $d = 2k, k \in \mathbb{N}$ and note that $\Gamma(1) = 1, \Gamma(\frac{1}{2}) = \pi^{\frac{1}{2}}$, then

$$\frac{\Gamma(\frac{d}{2}+1)}{\Gamma(\frac{d+1}{2})} = \frac{\Gamma(k+1)}{\Gamma(k+\frac{1}{2})} = \frac{k!\Gamma(1)}{\Pi_{i=1}^{k}(i-\frac{1}{2})\Gamma(\frac{1}{2})} = \frac{k!}{\pi^{\frac{1}{2}}\Pi_{i=1}^{k}(i-\frac{1}{2})} = \frac{(2k)!!}{\pi^{\frac{1}{2}}(2k-1)!!} \tag{102}$$

Recall the Wallis integral lemma that when $d$ is even

$$\int_0^{\frac{\pi}{2}} \sin^d(t)dt = \int_0^{\frac{\pi}{2}} \cos^d(t)dt = \frac{\pi}{2} \cdot \frac{(d-1)!!}{d!!}, \quad d = 2k \in \mathbb{N} \tag{103}$$

Thus,

$$\frac{\Gamma(\frac{d}{2}+1)}{\Gamma(\frac{d+1}{2})} = \frac{(2k)!!}{\pi^{\frac{1}{2}}(2k-1)!!} = \frac{1}{2\pi^{-\frac{1}{2}} \cdot (\frac{\pi}{2} \cdot \frac{(2k-1)!!}{(2k)!!})} = \frac{1}{2\pi^{-\frac{1}{2}} \int_0^{\frac{\pi}{2}} \sin^{2k}(t)dt} = \frac{1}{2\pi^{-\frac{1}{2}} \int_0^{\frac{\pi}{2}} \sin^d(t)dt} \tag{104}$$

When $d$ is odd, assume $d = 2k+1, k \in \mathbb{N}$ and note that $\Gamma(1) = 1, \Gamma(\frac{1}{2}) = \pi^{\frac{1}{2}}$, then

$$\frac{\Gamma(\frac{d}{2}+1)}{\Gamma(\frac{d+1}{2})} = \frac{\Gamma(k+\frac{3}{2})}{\Gamma(k+1)} = \frac{\Pi_{i=0}^{k}(i+\frac{1}{2})\Gamma(\frac{1}{2})}{k!\Gamma(1)} = \frac{\pi^{\frac{1}{2}}\Pi_{i=0}^{k}(i+\frac{1}{2})}{k!} = \frac{\pi^{\frac{1}{2}}(2k+1)!!}{2 \cdot (2k)!!} \tag{105}$$

Recall the Wallis integral lemma that when $d$ is odd

$$\int_0^{\frac{\pi}{2}} \sin^d(t)dt = \int_0^{\frac{\pi}{2}} \cos^d(t)dt = \frac{(d-1)!!}{d!!}, \quad d = 2k+1 \in \mathbb{N} \tag{106}$$

Thus,

$$\frac{\Gamma(\frac{d}{2}+1)}{\Gamma(\frac{d+1}{2})} = \frac{\pi^{\frac{1}{2}}(2k+1)!!}{2 \cdot (2k)!!} = \frac{1}{2\pi^{-\frac{1}{2}} \cdot (\frac{(2k)!!}{(2k+1)!!})} = \frac{1}{2\pi^{-\frac{1}{2}} \int_0^{\frac{\pi}{2}} \sin^{2k+1}(t)dt} = \frac{1}{2\pi^{-\frac{1}{2}} \int_0^{\frac{\pi}{2}} \sin^d(t)dt} \tag{107}$$

To sum up, for all $d \in \mathbb{N}$, we have

$$\frac{\Gamma(\frac{d}{2}+1)}{\Gamma(\frac{d+1}{2})} = \frac{1}{2\pi^{-\frac{1}{2}} \int_0^{\frac{\pi}{2}} \sin^d(t)dt} \tag{108}$$

Then we avoid the computation of $\Gamma(\frac{d}{2}+1), \Gamma(\frac{d+1}{2})$ and transfer it into the computation of an integral. Applying formula 108, we have

$$\sup_{||z||_q \leq \epsilon} TV(\mu, z+\mu) = 1 - \frac{1}{\int_0^{\frac{\pi}{2}} \sin^d(t)dt} \int_0^{\arccos(\frac{\max\{\epsilon,\epsilon d^{\frac{1}{2}-\frac{1}{q}}\}}{2r})} \sin^d(t)dt \tag{109}$$

$\square$

## H    PROOF OF THEOREM 3.5

*Proof.* Recall the definition of $l_p$-norm constraint set of probability measures

$$\mathcal{D}_{x,\epsilon,q} = \{x' + \mu : ||x - x'||_q \leq \epsilon\} \tag{110}$$

Note that

$$\sup_{\nu \in \mathcal{D}_{x,\epsilon,q}} TV(\mu, \nu) = \sup_{||x-x'||_q \leq \epsilon} TV(x+\mu, x'+\mu) = \sup_{||z||_q \leq \epsilon} TV(\mu, z+\mu) \tag{111}$$

where the first equality is due to the definition of $\mathcal{D}_{x,\epsilon,q}$ and the second equality is due to the translation invariance property of total variance distance.

When $\epsilon \geq 2r$,

$$1 \geq \sup_{||z||_q \leq \epsilon} TV(\mu, z+\mu) \geq TV(\mu, \epsilon e_1 + \mu) = 1 \tag{112}$$

where the first inequality is due to the fact that $\mu$ and $z + \mu$ are probability measures; the second inequality is due to $\text{supp}(\mu) \cap \text{supp}(\epsilon e_1 + \mu) = \emptyset$. Thus, in this case,

$$\sup_{||z||_q \leq \epsilon} TV(\mu, z + \mu) = 1 \tag{113}$$

When $\epsilon < 2r$,

$$\sup_{||z||_q \leq \epsilon} TV(\mu, z + \mu)$$

$$= \sup_{||z||_q \leq \epsilon} \frac{1}{2} \int \left| \frac{1}{\text{Vol}(K)} \mathcal{I}_{x \in K} - \frac{1}{2\text{Vol}(K)} \mathcal{I}_{x \in z+K} \right| dx \tag{114}$$

$$= \sup_{||z||_q \leq \epsilon} \frac{1}{2\text{Vol}(K)} \int \mathcal{I}_{x \in K \Delta (z+K)} dx = \sup_{||z||_q \leq \epsilon} \frac{\text{Vol}(K \Delta (z+K))}{2\text{Vol}(K)} = \sup_{||z||_q \leq \epsilon} 1 - \frac{\text{Vol}(K \cap (z+K))}{\text{Vol}(K)}$$

$$= \sup_{||z||_q \leq \epsilon} 1 - \frac{\Pi_{i=1}^d (2r - |z_i|)}{(2r)^d} = \sup_{||z||_q \leq \epsilon} 1 - \Pi_{i=1}^d \left(1 - \frac{|z_i|}{2r}\right) \tag{115}$$

First, we study typical cases when $q = 1, 2, \infty$. When $q = 1$, we need to solve the following optimization problem

$$\inf_{||z||_1 \leq \epsilon} \Pi_{i=1}^d (2r - |z_i|) \tag{116}$$

Here we use mathematical induction to prove that

$$\inf_{||z||_1 \leq \epsilon} \Pi_{i=1}^d (2r - |z_i|) = (2r)^{d-1}(2r - \epsilon) \tag{117}$$

When $d = 2$,

$$\inf_{||z||_1 \leq \epsilon} \Pi_{i=1}^d (2r - |z_i|) = \inf_{|z_1| + |z_2| \leq \epsilon} (2r - |z_1|)(2r - |z_2|) = \inf_{|z_2| \leq \epsilon} (2r - \epsilon + |z_2|)(2r - |z_2|)$$

$$= \inf_{0 \leq z_2 \leq \epsilon} (2r - \epsilon + z_2)(2r - z_2) = \inf_{0 \leq z_2 \leq \epsilon} z_2(\epsilon - z_2) + 2r(2r - \epsilon) = 2r(2r - \epsilon)$$

Thus, induction hypothesis holds for $d = 2$. Then, assume induction hypothesis holds for $d = n$. When $d = n + 1$,

$$\inf_{||z||_1 \leq \epsilon} \Pi_{i=1}^{n+1} (2r - |z_i|) = \inf_{\sum_{i=1}^{n+1} |z_i| \leq \epsilon} \Pi_{i=1}^{n+1} (2r - |z_i|) = \inf_{\sum_{i=1}^n |z_i| \leq \epsilon - |z_{n+1}|} \left(\Pi_{i=1}^n (2r - |z_i|)\right)(2r - |z_{n+1}|)$$

$$= \inf_{|z_{n+1}| \leq \epsilon} (2r)^{n-1}(2r - \epsilon + |z_{n+1}|)(2r - |z_{n+1}|) \tag{118}$$

$$= (2r)^n (2r - \epsilon) = (2r)^{d-1}(2r - \epsilon) \tag{119}$$

where 118 is due to the induction hypothesis when $d = n$; 119 is due to the induction hypothesis when $d = 2$. Therefore, we have already proved that

$$\inf_{||z||_1 \leq \epsilon} \Pi_{i=1}^d (2r - |z_i|) = (2r)^{d-1}(2r - \epsilon), \forall d \in \mathbb{N} \tag{120}$$

Plugging in this result, it follows that

$$\sup_{||z||_1 \leq \epsilon} TV(\mu, z + \mu) = \sup_{||z||_1 \leq \epsilon} 1 - \frac{\Pi_{i=1}^d (2r - |z_i|)}{(2r)^d} = 1 - \frac{(2r)^{d-1}(2r - \epsilon)}{(2r)^d} = \frac{\epsilon}{2r} \tag{121}$$

When $q = 2$, we need to solve the following optimization problem

$$\inf_{||z||_2 \leq \epsilon} \Pi_{i=1}^d (2r - |z_i|) \tag{122}$$

When $d = 2$,

$$\inf_{||z||_2 \leq \epsilon} \Pi_{i=1}^d (2r - |z_i|) = \inf_{|z_1|^2 + |z_2|^2 \leq \epsilon^2} (2r - |z_1|)(2r - |z_2|) = \inf_{|z_2| \leq \epsilon} \left(2r - (\epsilon^2 - z_2^2)^{\frac{1}{2}}\right)(2r - |z_2|)$$

$$= \inf_{0 \leq z_2 \leq \epsilon} \left(2r - (\epsilon^2 - z_2^2)^{\frac{1}{2}}\right)(2r - z_2)$$

Define $f(z_2) = \ln\left(2r - (\epsilon^2 - z_2^2)^{\frac{1}{2}}\right) + \ln(2r - z_2)$, then

$$f'(z_2) = \frac{z_2(\epsilon^2 - z_2^2)^{-\frac{1}{2}}}{2r - (\epsilon^2 - z_2^2)^{\frac{1}{2}}} - \frac{1}{2r - z_2} = \frac{2rz_2(\epsilon^2 - z_2^2)^{-\frac{1}{2}} - z_2^2(\epsilon^2 - z_2^2)^{-\frac{1}{2}} - 2r + (\epsilon^2 - z_2^2)^{\frac{1}{2}}}{\left(2r - (\epsilon^2 - z_2^2)^{\frac{1}{2}}\right)(2r - z_2)}$$
(123)

Define $g(z_2) = 2rz_2(\epsilon^2 - z_2^2)^{-\frac{1}{2}} - z_2^2(\epsilon^2 - z_2^2)^{-\frac{1}{2}} - 2r + (\epsilon^2 - z_2^2)^{\frac{1}{2}}$, then

$$g'(z_2) = (2z_2^3 - 3\epsilon^2 z_2 + 2r\epsilon^2)(\epsilon^2 - z_2^2)^{-\frac{3}{2}}$$
(124)

Define $h(z_2) = 2z_2^3 - 3\epsilon^2 z_2 + 2r\epsilon^2$, then $h'(z_2) = 6z_2^2 - 3\epsilon^2 = 6(z_2 - \frac{\epsilon}{\sqrt{2}})(z_2 + \frac{\epsilon}{\sqrt{2}})$. Thus, when $0 \le z_2 \le \frac{\epsilon}{\sqrt{2}}$, $h'(x) \le 0$; when $\frac{\epsilon}{\sqrt{2}} < z_2 \le \epsilon$, $h'(x) > 0$. Thus, the minimum value of $h(x)$ on interval $[0, \epsilon]$ is $h(\frac{\epsilon}{\sqrt{2}}) = \sqrt{2}\epsilon^2(\sqrt{2}r - \epsilon)$. Therefore, function $f(z_2)$ behaves differently when $0 < \epsilon \le \sqrt{2}r$ and when $\sqrt{2}r < \epsilon < 2r$.

When $0 < \epsilon \le \sqrt{2}r$, $h(z_2) \ge h(\frac{\epsilon}{\sqrt{2}}) = \sqrt{2}\epsilon^2(\sqrt{2}r - \epsilon) \ge 0$ on interval $[0, \epsilon]$ and therefore $g'(z_2) = h(z_2)(\epsilon^2 - z_2^2)^{-\frac{3}{2}} \ge 0$. Note that $g(0) = \epsilon - 2r < 0, g(\frac{\epsilon}{\sqrt{2}}) = 0, g(\epsilon^-) = \infty$ and therefore $f'(z_2) \le 0$ when $0 \le z_2 \le \frac{\epsilon}{\sqrt{2}}$ while $f'(z_2) > 0$ when $\frac{\epsilon}{\sqrt{2}} < z_2 \le \epsilon$. Thus, $f(z_2)$ takes its minimum when $z_2 = \frac{\epsilon}{\sqrt{2}}$. In this case,

$$\inf_{0 \le z_2 \le \epsilon} \left(2r - (\epsilon^2 - z_2^2)^{\frac{1}{2}}\right)(2r - z_2) = (2r - \frac{\epsilon}{\sqrt{2}})^2$$
(125)

When $\sqrt{2}r < \epsilon < 2r$, we have $h(0) = 2r\epsilon^2 > 0, h(\frac{\epsilon}{\sqrt{2}}) = \sqrt{2}\epsilon^2(\sqrt{2}r - \epsilon) < 0, h(\epsilon) = \epsilon^2(2r - \epsilon) > 0$. Assume $h(t_1) = h(t_2) = 0, 0 < t_1 < \frac{\epsilon}{\sqrt{2}} < t_2 < \epsilon$, then when $0 \le z_2 \le t_1$ or $t_2 \le z_2 \le \epsilon$, $h(z_2) \ge 0$ and when $t_1 < z_2 < t_2$, $h(z_2) < 0$. Therefore, $g'(z_2) \ge 0$ when $0 \le z_2 \le t_1$ or $t_2 \le z_2 \le \epsilon$; $g'(z_2) < 0$ when $t_1 < z_2 < t_2$. Note that $g(z_2) = 0 \iff (2z_2^2 - \epsilon^2)(2(z_2 - r)^2 + 2r^2 - \epsilon^2) = 0$, therefore when $0 \le z_2 \le r - \sqrt{\frac{\epsilon^2}{2} - r^2}$ or $\frac{\epsilon}{\sqrt{2}} \le z_2 \le r + \sqrt{\frac{\epsilon^2}{2} - r^2}$, $g(z_2) \le 0$; when $r - \sqrt{\frac{\epsilon^2}{2} - r^2} < z_2 < \frac{\epsilon}{\sqrt{2}}$ or $r + \sqrt{\frac{\epsilon^2}{2} - r^2} < z_2 < \epsilon$, $g(z_2) > 0$. Thus, when $0 \le z_2 \le r - \sqrt{\frac{\epsilon^2}{2} - r^2}$ or $\frac{\epsilon}{\sqrt{2}} \le z_2 \le r + \sqrt{\frac{\epsilon^2}{2} - r^2}$, $f'(x) \le 0$; when $r - \sqrt{\frac{\epsilon^2}{2} - r^2} < z_2 < \frac{\epsilon}{\sqrt{2}}$ or $r + \sqrt{\frac{\epsilon^2}{2} - r^2} < z_2 < \epsilon$, $f'(x) > 0$. Thus, $f(z_2)$ takes its minimum when $z_2 = r - \sqrt{\frac{\epsilon^2}{2} - r^2}$ or $z_2 = r + \sqrt{\frac{\epsilon^2}{2} - r^2}$. In this case,

$$\inf_{0 \le z_2 \le \epsilon} \left(2r - (\epsilon^2 - z_2^2)^{\frac{1}{2}}\right)(2r - z_2) = \left(r - \sqrt{\frac{\epsilon^2}{2} - r^2}\right)\left(r + \sqrt{\frac{\epsilon^2}{2} - r^2}\right) = 2r^2 - \frac{\epsilon^2}{2}$$
(126)

$$\inf_{||z||_2 \le \epsilon} \Pi_{i=1}^d (2r - |z_i|) = \inf_{\sum_{i=1}^{n+1} z_i^2 \le \epsilon^2} \Pi_{i=1}^{n+1}(2r - |z_i|) = \inf_{\sum_{i=1}^n z_i^2 \le \epsilon^2 - z_{n+1}^2} \left(\Pi_{i=1}^n (2r - |z_i|)\right)(2r - |z_{n+1}|)$$
(127)

By then, we have understand clearly the optimization problem when $d = 2$.

Then, consider the case when $d = 3$. When $d = 3$,

$$\inf_{||z||_2 \le \epsilon} \Pi_{i=1}^d (2r - |z_i|) = \inf_{z_1^2 + z_2^2 + z_3^2 \le \epsilon^2} (2r - |z_1|)(2r - |z_2|)(2r - |z_3|)$$
(128)

When $0 < \epsilon \le \sqrt{2}r$, assume the optimal point is $z^*$. We will prove that each coordinate of $z^*$ has the same value. Here we use reduction to absurdity, and wlog assume $z_1^* \ne z_2^*$. By fixing the value of $z_3^*$, the optimization problem 122 is equivalent to

$$\inf_{z_1^2 + z_2^2 \le \epsilon^2 - (z_3^*)^2} (2r - |z_1|)(2r - |z_2|)$$
(129)

And $(z_1^*, z_2^*)$ should be an optimal point of above problem. Note that $\epsilon^2 - (z_3^*)^2 \le \epsilon^2 \le 2r^2$ and applying 125, we know that $z_1^* = z_2^*$ which is a contradiction. Thus, $z_1^* = z_2^* = z_3^* = c$. And

$$\inf_{||z||_2 \le \epsilon} \Pi_{i=1}^n (2r - |z_i|) = \inf_{c \le \frac{\epsilon}{\sqrt{3}}} (2r - c)^3 = \left(2r - \frac{\epsilon}{\sqrt{3}}\right)^3$$
(130)

When $\sqrt{2}r < \epsilon \le \sqrt{3}r$, it's obvious that the optimal point $z^*$ of optimization problem 128 must lie on the boundary of feasible region, i.e. $(z_1^*)^3 + (z_2^*)^3 + (z_3^*)^3 = \epsilon^2$. Wlog, assume $(z_3^*)^3 \ge \frac{\epsilon^2}{3}$ and $(z_1^*)^2 + (z_2^*)^2 \le \frac{2\epsilon^2}{3} \le 2r^3$. By fixing the value of $z_3^*$ and following similar deduction procedure as above we know that $z_1^* = z_2^* = c^*$, where $c^*$ is the optimal point of following optimization problem.

$$\inf_{0 \le c \le \frac{\epsilon}{\sqrt{3}}} (2r - c)^2 (2r - \sqrt{\epsilon^2 - 2c^2}) \tag{131}$$

Define $f(x) = 2\ln(2r - x) + \ln(2r - \sqrt{\epsilon^2 - 2x^2})$ where $0 \le x \le \frac{\epsilon}{\sqrt{3}}$, then

$$f'(x) = \frac{2(3x^2 - 2rx - \epsilon^2 + 2r\sqrt{\epsilon^2 - 2x^2})}{(x - 2r)(2r - \sqrt{\epsilon^2 - 2x^2})\sqrt{\epsilon^2 - 2x^2}} \tag{132}$$

It's obvious that the denominator of $f'(x)$ is negative. As for the numerator, define $g(x) = 3x^2 - 2rx - \epsilon^2$ where $0 \le x \le \frac{\epsilon}{\sqrt{3}}$. Note that

$$g(x) \le \max\left\{ g(0), g\left(\frac{\epsilon}{\sqrt{3}}\right) \right\} = \max\left\{ -\epsilon^2, -\frac{2r\epsilon}{\sqrt{3}} \right\} \le 0 \tag{133}$$

Thus, we have the following equivalent relationship

$$3x^2 - 2rx - \epsilon^2 + 2r\sqrt{\epsilon^2 - 2x^2} \le 0$$
$$\Longleftrightarrow 3x^2 - 2rx - \epsilon^2 \le -2r\sqrt{\epsilon^2 - 2x^2} \le 0$$
$$\Longleftrightarrow (3x^2 - 2rx - \epsilon^2)^2 \ge \left(-2r\sqrt{\epsilon^2 - 2x^2}\right)^2 \ge 0$$
$$\Longleftrightarrow (3x^2 - \epsilon^2)(3x^2 - 4rx + 4r^2 - \epsilon^2) \ge 0$$
$$\Longleftrightarrow 3x^2 - 4rx + 4r^2 - \epsilon^2 \le 0$$
$$\Longleftrightarrow \begin{cases} \emptyset & \text{when } \sqrt{2}r < \epsilon \le 2\sqrt{\frac{2}{3}}r \\ \dfrac{2r - \sqrt{3\epsilon^2 - 8r^2}}{3} \le x \le \dfrac{2r + \sqrt{3\epsilon^2 - 8r^2}}{3} & \text{when } 2\sqrt{\frac{2}{3}}r < \epsilon \le \sqrt{3}r \end{cases}$$

where the last equivalent relationship is due to the discriminant of the quadratic equation $3x^2 - 4rx + 4r^2 - \epsilon^2$ is $\Delta = 4(3\epsilon^2 - 8r^2)$. Therefore, when $\sqrt{2}r < \epsilon \le 2\sqrt{\frac{2}{3}}r$, $f'(x) \le 0, \forall 0 \le x \le \frac{\epsilon}{\sqrt{3}}$ and hence the optimal point $c^*$ in the optimization problem 131 takes value $\frac{\epsilon}{\sqrt{3}}$, whereas when $2\sqrt{\frac{2}{3}}r < \epsilon \le \sqrt{3}r$, $f'(x) \le 0$ for $0 \le x \le \frac{2r - \sqrt{3\epsilon^2 - 8r^2}}{3}$, $\frac{2r + \sqrt{3\epsilon^2 - 8r^2}}{3} \le x \le \frac{\epsilon}{\sqrt{3}}$ and $f'(x) > 0$ for $\frac{2r - \sqrt{3\epsilon^2 - 8r^2}}{3} \le x \le \frac{2r + \sqrt{3\epsilon^2 - 8r^2}}{3}$ and note that $f(\frac{2r - \sqrt{3\epsilon^2 - 8r^2}}{3}) < f(\frac{\epsilon}{\sqrt{3}})$ hence the optimal point $c^*$ in the optimization problem takes value $\frac{2r - \sqrt{3\epsilon^2 - 8r^2}}{3}$. To sum up, when $\sqrt{2}r < \epsilon \le 2\sqrt{\frac{2}{3}}r$, the optimal point $z^*$ of optimization problem 128 satisfies $z_1^* = z_2^* = z_3^* = \frac{\epsilon}{\sqrt{3}}$. And when $2\sqrt{\frac{2}{3}}r < \epsilon \le \sqrt{3}r$, the optimal point $z^*$ of optimization problem 128 satisfies $z_1^* = z_2^* = \frac{2r - \sqrt{3\epsilon^2 - 8r^2}}{3}, z_3^* = \frac{4r + \sqrt{3\epsilon^2 - 8r^2}}{3}$ or one of its permutations.

When $\sqrt{3}r < \epsilon < 2r$, similarly we have $(z_1^*)^3 + (z_2^*)^3 + (z_3^*)^3 = \epsilon^2$. If there exists $1 \le i \le 3$ such that $(z_i^*)^2 \ge \epsilon^2 - 2r^2$, wlog assume $(z_3^*)^2 \ge \epsilon^2 - 2r^2$. By substituting the value range of $x$ from $[0, \frac{\epsilon}{\sqrt{3}}]$ into $[0, r]$, following similar deduction procedure and noticing that $f(\frac{2r - \sqrt{3\epsilon^2 - 8r^2}}{3}) < f(r)$, we know that the optimal point $z^*$ in this case satisfies $z_1^* = z_2^* = \frac{2r - \sqrt{3\epsilon^2 - 8r^2}}{3}, z_3^* = \frac{4r + \sqrt{3\epsilon^2 - 8r^2}}{3}$ or one of its permutations. On the other hand, if $(z_i^*)^2 < \epsilon^2 - 2r^2$ for all $1 \le i \le 3$, then $(z_1^*)^2 + (z_2^*)^2 = \epsilon^2 - (z_3^*)^2 > 2r^2$. Applying 126 and taking $z_1^* = r - \sqrt{\frac{\epsilon^2 - (z_3^*)^2}{2} - r^2}, z_2^* = r + \sqrt{\frac{\epsilon^2 - (z_3^*)^2}{2} - r^2}$, we know the optimization problem is equivalent to

$$\inf_{0 \le z_3 < \sqrt{\epsilon^2 - 2r^2}} \left(2r^2 - \frac{\epsilon^2 - z_3^2}{2}\right)(2r - z_3) \tag{134}$$

According to monotonicity analysis of the cubic function above, the optimal point $z_3^*$ is either $\frac{2r - \sqrt{3\epsilon^2 - 8r^2}}{3}$ or $\sqrt{\epsilon^2 - 2r^2}$. And it's easy to verify that $f\left(\frac{2r - \sqrt{3\epsilon^2 - 8r^2}}{3}\right) < f\left(\sqrt{\epsilon^2 - 2r^2}\right)$ and therefore $z_3^* = \frac{2r - \sqrt{3\epsilon^2 - 8r^2}}{3}$. However, $(z_2^*)^2 > \epsilon^2 - 2r^2$ which leads to a contradiction.

In summary, considering the case $d = 3$, when $0 < \epsilon \le 2\sqrt{\frac{2}{3}}r$, the optimal point $z^*$ of original optimization problem satisfies $z_1^* = z_2^* = z_3^* = \frac{\epsilon}{\sqrt{3}}$ and the optimal value is $(2r - \frac{\epsilon}{\sqrt{3}})^3$ and when $\sqrt{3}r < \epsilon < 2r$, the optimal point $z^*$, the optimal point $z^*$ of original optimization problem satisfies $z_1^* = z_2^* = \frac{2r - \sqrt{3\epsilon^2 - 8r^2}}{3}, z_3^* = \frac{4r + \sqrt{3\epsilon^2 - 8r^2}}{3}$ or one of its permutations.

Next, consider the general case when $d = n \ge 4$. In the first place, we point out and prove two useful properties of the optimal point $z^*$ which help simplify our later discussion a lot.

- All coordinates of optimal point $z^*$ takes at most two different values.

- If the coordinates of an optimal point $z^*$ takes exactly two different values $c_1$ and $c_2$, then the number of coordinates equal to $c_1$ must be $n - 1$ or 1.

*Proof.* On one hand, by using reduction to absurdity, wlog assume $z_1^*, z_2^*, z_3^*$ take three different values. Fixing the value of the other $n - 3$ coordinates, we know that $(z_1^*, z_2^*, z_3^*)$ is the optimal point of a special case of original problem when $d = 3$. And note that for all the optimal points of $d = 3$, there must exist two coordinates taking the same value, which leads to a contradiction. Thus, the first property is satisfied.
On the other hand, similarly, by applying reduction to absurdity, wlog assume $z_1^* = z_2^* = c_1$ and $z_3^* = z_4^* = c_2$ where $c_1 \ne c_2$. Fixing $z_2^*, z_4^*$ and the value of the other $n - 4$ coordinates and aware of the fact that $(z_1^*)^2 + (z_3^*)^2 \le \frac{\epsilon^2}{2} < 2r^2$, we know that $(z_1^*, z_3^*)$ is the optimal point of a special case of original problem when $d = 2, \epsilon < \sqrt{2}r$ and therefore $z_1^* = z_3^*$, which leads to a contradiction. Thus, the second property is satisfied. □

Using the two properties above, we know that the optimal point $z^*$ has only two possible forms: $z^* = \left(\frac{\epsilon}{\sqrt{n}}, \cdots, \frac{\epsilon}{\sqrt{n}}\right)$ and $z^* = \left(c, \cdots, c, \sqrt{\epsilon^2 - (n-1)c^2}\right)$ or one of its permutations where $0 \le c \le \frac{\epsilon}{\sqrt{n-1}}, c \ne \frac{\epsilon}{\sqrt{n}}$, which can be unified into one form: $z^* = \left(c, \cdots, c, \sqrt{\epsilon^2 - (n-1)c^2}\right)$ or one of its permutations where $0 \le c \le \frac{\epsilon}{\sqrt{n-1}}$. Thus, the original problem can be simplified into following optimization problem with one degree of freedom:

$$\inf_{0 \le c \le \frac{\epsilon}{\sqrt{n-1}}} (2r - c)^{n-1}\left(2r - \sqrt{\epsilon^2 - (n-1)c^2}\right) \tag{135}$$

Define $f(x) = (n-1)\ln(2r - x) + \ln\left(2r - \sqrt{\epsilon^2 - (n-1)x^2}\right)$ where $0 \le x \le \frac{\epsilon}{\sqrt{n-1}}$, then

$$f'(x) = \frac{(n-1)\left(nx^2 - 2rx - \epsilon^2 + 2r\sqrt{\epsilon^2 - (n-1)x^2}\right)}{(x - 2r)\left(2r - \sqrt{\epsilon^2 - (n-1)x^2}\right)\sqrt{\epsilon^2 - (n-1)x^2}}, \text{ where } 0 \le x < \frac{\epsilon}{\sqrt{n-1}} \tag{136}$$

It's obvious that the denominator of $f'(x)$ is negative. As for the numerator, define $g(x) = nx^2 - 2rx - \epsilon^2$ where $0 \le x \le \frac{\epsilon}{\sqrt{n-1}}$. Note that

$$g(x) \le \min\left\{g(0), g\left(\frac{\epsilon}{\sqrt{n-1}}\right)\right\} = \max\left\{0, \frac{\epsilon(\epsilon - 2\sqrt{n-1}r)}{n-1}\right\} \le 0 \tag{137}$$

where the last inequality is due to $\epsilon < 2r < 2\sqrt{n-1}r$. Thus, when $0 \le x \le \frac{\epsilon}{\sqrt{n-1}}$,

$$nx^2 - 2rx - \epsilon^2 + 2r\sqrt{\epsilon^2 - (n-1)x^2} \le 0$$
$$\iff nx^2 - 2rx - \epsilon^2 \le -2r\sqrt{\epsilon^2 - (n-1)x^2} \le 0$$
$$\iff (nx^2 - 2rx - \epsilon^2)^2 \ge \left(-2r\sqrt{\epsilon^2 - (n-1)x^2}\right)^2$$
$$\iff (nx^2 - \epsilon^2)(nx^2 - 4rx + 4r^2 - \epsilon^2) \ge 0$$
$$\iff \left(x - \frac{\epsilon}{\sqrt{n}}\right)(nx^2 - 4rx + 4r^2 - \epsilon^2) \ge 0$$
$$\iff \begin{cases} x \ge \frac{\epsilon}{\sqrt{n}} & \text{when } 0 < \epsilon < 2\sqrt{\frac{n-1}{n}}r \\[2ex] x \ge \frac{\epsilon}{\sqrt{n}} \text{ or } \frac{2r - \sqrt{n\epsilon^2 - 4(n-1)r^2}}{n} \le x \le \frac{2r + \sqrt{n\epsilon^2 - 4(n-1)r^2}}{n} & \text{when } 2\sqrt{\frac{n-1}{n}}r \le \epsilon < 2r \end{cases}$$

where the last equivalence relationship is due to the discriminant of the quadratic equation $nx^2 - 4rx + 4r^2 - \epsilon^2 = 0$ is

$$\Delta = 4\left(n\epsilon^2 - 4(n-1)r^2\right) < 0 \iff 0 < \epsilon < 2\sqrt{\frac{n-1}{n}}r \tag{138}$$

Thus, if $0 < \epsilon < 2\sqrt{\frac{n-1}{n}}r$, then $f'(x) \ge 0$ when $\frac{\epsilon}{\sqrt{n}} \le x \le \frac{\epsilon}{\sqrt{n-1}}$ and $f'(x) < 0$ when $0 \le x < \frac{\epsilon}{\sqrt{n}}$. Thus, $f(x)$ takes its minimum when $x = \frac{\epsilon}{\sqrt{n}}$ and therefore $c^* = \frac{\epsilon}{\sqrt{n}}$.

If $2\sqrt{\frac{n-1}{n}}r \le \epsilon < 2r$, then $f'(x) \ge 0$ when $\frac{2r - \sqrt{n\epsilon^2 - 4(n-1)r^2}}{n} \le x \le \frac{2r + \sqrt{n\epsilon^2 - 4(n-1)r^2}}{n}$ or $\frac{\epsilon}{\sqrt{n}} \le x \le \frac{\epsilon}{\sqrt{n-1}}$ and $f'(x) < 0$ when $0 \le x < \frac{2r - \sqrt{n\epsilon^2 - 4(n-1)r^2}}{n}$ or $\frac{2r + \sqrt{n\epsilon^2 - 4(n-1)r^2}}{n} < x < \frac{\epsilon}{\sqrt{n}}$. In this case, $f(x)$ takes its minimum when $x = \frac{2r - \sqrt{n\epsilon^2 - 4(n-1)r^2}}{n}$ or $x = \frac{\epsilon}{\sqrt{n}}$. For the convenience of analysis, assume $t = \frac{\epsilon}{2r}$, $\sqrt{\frac{n-1}{n}} \le t < 1$ and it follows that

$$e^{f(\frac{\epsilon}{\sqrt{n}})} = (2r)^n \left(1 - \frac{t}{\sqrt{n}}\right)^n \tag{139}$$

$$e^{f\left(\frac{2r - \sqrt{n\epsilon^2 - 4(n-1)r^2}}{n}\right)} = (2r)^n \left(\frac{(n-1) + \sqrt{nt^2 - (n-1)}}{n}\right)^{n-1} \left(\frac{1 - \sqrt{nt^2 - (n-1)}}{n}\right) \tag{140}$$

We can prove that there exists $t_n \in \left[\sqrt{\frac{n-1}{n}}, 1\right)$ such that $c^* = \frac{\epsilon}{\sqrt{n}}$ when $2\sqrt{\frac{n-1}{n}}r \le \epsilon \le 2t_n r$ and $c^* = \frac{2r - \sqrt{n\epsilon^2 - 4(n-1)r^2}}{n}$ when $2t_n r < \epsilon < 2r$ while $t_n$ converge to $1$ at an exponential rate as shown in figure 5.

In conclusion, for the case $d = n \ge 4$, when $0 < \epsilon \le 2t_n r$, $c^* = \frac{\epsilon}{\sqrt{n}}$ and therefore

$$\inf_{||z||_2 \le \epsilon} \Pi_{i=1}^n (2r - |z_i|) = (2r - c^*)^{n-1}\left(2r - \sqrt{\epsilon^2 - (n-1)(c^*)^2}\right) = \left(2r - \frac{\epsilon}{n^{\frac{1}{2}}}\right)^n = \left(2r - \frac{\epsilon}{d^{\frac{1}{2}}}\right)^d \tag{141}$$

Plugging in this result, it follows that

$$\sup_{||z||_2 \le \epsilon} TV(\mu, z + \mu) = \sup_{||z||_2 \le \epsilon} 1 - \frac{\Pi_{i=1}^d (2r - |z_i|)}{(2r)^d} = 1 - \frac{\left(2r - \frac{\epsilon}{d^{\frac{1}{2}}}\right)^d}{(2r)^d} = 1 - \left(1 - \frac{\epsilon}{2d^{\frac{1}{2}}r}\right)^d \tag{142}$$

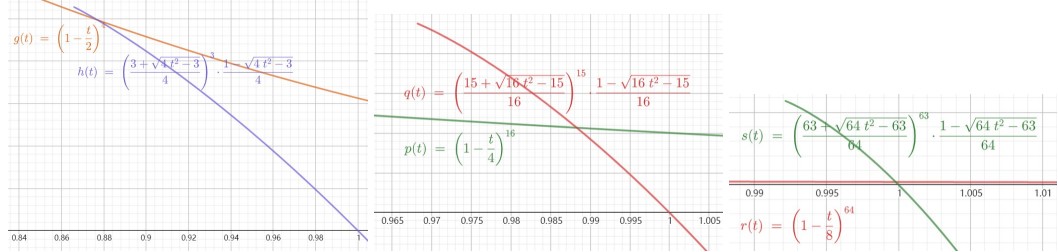

Figure 5: Graphs of functions $f_1(t) = \left(1 - \frac{t}{\sqrt{n}}\right)^n$, $f_2(t) = \left(\frac{(n-1)+\sqrt{nt^2-(n-1)}}{n}\right)^{n-1}\left(\frac{1-\sqrt{nt^2-(n-1)}}{n}\right)$ when $n = 4, 16, 64$ from left to right. According to the figure, on interval $\left[\sqrt{\frac{n-1}{n}}, 1\right]$, $f_2(t)$ is greater than $f_1(t)$ at first and then $f_2(t)$ exceeds $f_1(t)$. Furthermore, as $n$ increases, the horizontal coordinate of the intersection point converge to 1, which can be seen intuitively from the figure above.

and when $2t_n r < \epsilon < 2r$, $c^* = \frac{2r-\sqrt{n\epsilon^2-4(n-1)r^2}}{n}$ and therefore

$$\inf_{||z||_2 \leq \epsilon} \Pi_{i=1}^n (2r - |z_i|) = (2r - c^*)^{n-1}\left(2r - \sqrt{\epsilon^2 - (n-1)(c^*)^2}\right)$$

$$= \left(\frac{2(n-1)r + \sqrt{n\epsilon^2 - 4(n-1)r^2}}{n}\right)^{n-1}\left(\frac{2r - \sqrt{n\epsilon^2 - 4(n-1)r^2}}{n}\right)$$

$$= \left(\frac{2(d-1)r + \sqrt{d\epsilon^2 - 4(d-1)r^2}}{n}\right)^{n-1}\left(\frac{2r - \sqrt{d\epsilon^2 - 4(d-1)r^2}}{d}\right)$$

Plugging in this result, it follows that

$$\sup_{||z||_2 \leq \epsilon} TV(\mu, z + \mu) = \sup_{||z||_2 \leq \epsilon} 1 - \frac{\Pi_{i=1}^d (2r - |z_i|)}{(2r)^d}$$

$$= 1 - \frac{\left(\frac{2(d-1)r+\sqrt{d\epsilon^2-4(d-1)r^2}}{d}\right)^{d-1}\left(\frac{2r-\sqrt{d\epsilon^2-4(d-1)r^2}}{d}\right)}{(2r)^d}$$

$$= 1 - \left(\frac{d-1+\sqrt{d(\frac{\epsilon}{2r})^2 - d + 1}}{d}\right)^{d-1}\left(\frac{1-\sqrt{d(\frac{\epsilon}{2r})^2 - d + 1}}{d}\right)$$

When $q = \infty$, it's easy to verify that

$$\inf_{||z||_\infty \leq \epsilon} \Pi_{i=1}^d (2r - |z_i|) = (2r - \epsilon)^d \tag{143}$$

Plugging in the formula above, it follows

$$\sup_{||z||_\infty \leq \epsilon} TV(\mu, z + \mu) = \sup_{||z||_\infty \leq \epsilon} 1 - \frac{\Pi_{i=1}^d (2r - |z_i|)}{(2r)^d} = 1 - \frac{(2r - \epsilon)^d}{(2r)^d} = 1 - \left(1 - \frac{\epsilon}{2r}\right)^d \tag{144}$$

$\square$

## I   PROOF OF THEOREM 3.6

*Proof.* Recall the definition of $l_p$-norm constraint set of probability measures

$$\mathcal{D}_{x,\epsilon,q} = \{x' + \mu : ||x - x'||_q \leq \epsilon\} \tag{145}$$

Assume $\mathcal{D}_{x,\epsilon,q} \subseteq \mathcal{D}_{x,\xi(\epsilon)}$, then

$$\xi(\epsilon) \geq \sup_{\nu \in \mathcal{D}_{x,\epsilon,q}} TV(\mu, \nu) = \sup_{||x-x'||_q \leq \epsilon} TV(x+\mu, x'+\mu) = \sup_{||z||_q \leq \epsilon} TV(\mu, z+\mu) \geq TV(\mu, \epsilon e_1 + \mu) \tag{146}$$

which indicates that $TV(\mu, \epsilon e_1 + \mu)$ provides a lower bound for $\xi(\epsilon)$. Thus, we only need to estimate the value of $TV(\mu, \epsilon e_1 + \mu)$. According to lemma A.3, we have

$$TV(\mu, \epsilon e_1 + \mu) = \frac{\text{Vol}(K\Delta(\epsilon e_1 + K))}{2\text{Vol}(K)} = 1 - \frac{\text{Vol}(K \cap (\epsilon e_1 + K))}{\text{Vol}(K)}$$

Note that

$$K \cap (\epsilon e_1 + K)$$
$$= \left\{ x \in \mathbb{R}^d \Big| |x_1|^p + \cdots + |x_d|^p \leq r^p, |x_1 - \epsilon|^p + |x_2|^p + \cdots + |x_d|^p \leq r^p \right\}$$
$$= \left\{ x \in \mathbb{R}^d \Big| \epsilon - (r^p - (|x_2|^p + \cdots + |x_d|^p))^{\frac{1}{p}} \leq x_1 \leq (r^p - (|x_2|^p + \cdots + |x_d|^p))^{\frac{1}{p}} \right\}$$
$$= \left\{ x \in \mathbb{R}^d \Big| \epsilon - (r^p - (|x_2|^p + \cdots + |x_d|^p))^{\frac{1}{p}} \leq x_1 \leq \frac{\epsilon}{2} \right\} \cup \left\{ x \Big| \frac{\epsilon}{2} \leq x_1 \leq (r^p - (|x_2|^p + \cdots + |x_d|^p))^{\frac{1}{p}} \right\}$$
$$:= \Omega_1 \cup \Omega_2 \text{ where } \Omega_1 \cap \Omega_2 = \emptyset$$

It's easy to verify that $\text{Vol}(\Omega_1) = \text{Vol}(\Omega_2)$ according to integration by substitution and therefore $\text{Vol}(K \cap (\epsilon e_1 + K)) = 2\text{Vol}(\Omega_2)$. To estimate the volume of $\Omega_2$, we first introduce several lemmas below for the convenience of later discussion.

**Lemma I.1** (Volume formula of $d$-dimensional $l_p$ norm ball). *The volume of $d$-dimensional $l_p$ ball of radius $r$ is given by*

$$V_p^{(d)} = (2r)^d \frac{\Gamma(1 + \frac{1}{p})^d}{\Gamma(1 + \frac{d}{p})} \tag{147}$$

**Lemma I.2.** *The $d$-dimensional $l_p$ ball of volume 1 has radius about $\frac{d^{\frac{1}{p}}}{2(pe)^{\frac{1}{p}}\Gamma(1 + \frac{1}{p})}$.*

*Proof.* When dimension $d$ is big enough, we can obtain an asymptotic volume estimation of $l_p$ norm ball with radius $r$.

$$V_p^{(d)} = (2r)^d \frac{\Gamma(1 + \frac{1}{p})^d}{\Gamma(1 + \frac{d}{p})} \approx (2r)^d \frac{\Gamma(1 + \frac{1}{p})^d}{\sqrt{2\pi\frac{d}{p}}\left(\frac{d}{pe}\right)^{\frac{d}{p}}} = \sqrt{\frac{p}{2\pi d}}\left(\frac{2r(pe)^{\frac{1}{p}}\Gamma(1 + \frac{1}{p})}{d^{\frac{1}{p}}}\right)^d \tag{148}$$

where the first equality is due to lemma I.1 and the approximate equality is due to Stirling's formula about the estimation of gamma function that $\Gamma(z+1) \approx \sqrt{2\pi z}\left(\frac{z}{e}\right)^z$. Thus, when $V_p^{(d)} = 1$, we have

$$r \approx \frac{d^{\frac{1}{p}}}{2(pe)^{\frac{1}{p}}\Gamma(1 + \frac{1}{p})} \tag{149}$$

$\square$

Then we estimate the volume of $l_p$ norm ball cap by studying the asymptotic property of the mass distribution of $l_p$ norm ball. To begin with, let's estimate the $(d-1)$-dimensional volume of a slice through the center of the $l_p$ ball of volume 1. Note that the ball has radius $r = (V_p^{(d)})^{-\frac{1}{d}}$. The slice is an $(d-1)$-dimensional ball of this radius, so its volume is

$$V_p^{(d-1)} r^{d-1} = V_p^{(d-1)}(V_p^{(d)})^{-\frac{d-1}{d}} = 2^{d-1} \frac{\Gamma(1 + \frac{1}{p})^{d-1}}{\Gamma(1 + \frac{d-1}{p})}\left(2^d \frac{\Gamma(1 + \frac{1}{p})^d}{\Gamma(1 + \frac{d}{p})}\right)^{-\frac{d-1}{d}} \tag{150}$$

Using Stirling's formula again, when $d$ is sufficiently large, we have

$$V_p^{(d-1)} r^{d-1}$$
$$= 2^{d-1} \frac{\Gamma(1 + \frac{1}{p})^{d-1}}{\Gamma(1 + \frac{d-1}{p})}\left(2^d \frac{\Gamma(1 + \frac{1}{p})^d}{\Gamma(1 + \frac{d}{p})}\right)^{-\frac{d-1}{d}} = 2^{d-1} \frac{\Gamma(1 + \frac{1}{p})^{d-1}}{\sqrt{2\pi\frac{d-1}{p}}\left(\frac{d-1}{pe}\right)^{\frac{d-1}{p}}}\left(2^d \frac{\Gamma(1 + \frac{1}{p})^d}{\sqrt{2\pi\frac{d}{p}}\left(\frac{d}{pe}\right)^{\frac{d}{p}}}\right)^{-\frac{d-1}{d}}$$
$$= \frac{1}{\sqrt{2\pi\frac{d-1}{p}}\left(\frac{d-1}{pe}\right)^{\frac{d-1}{p}}} \cdot \frac{1}{\left(\sqrt{2\pi\frac{d}{p}}\left(\frac{d}{pe}\right)^{\frac{d}{p}}\right)^{-\frac{d-1}{d}}} = \frac{1}{\left(\frac{2\pi}{p}\right)^{\frac{1}{2d}}(d-1)^{\frac{d-1}{p}+\frac{1}{2}}d^{-\frac{d-1}{p}-\frac{d-1}{2d}}} = \frac{\left(1 + \frac{1}{d-1}\right)^{\frac{d-1}{p}+\frac{1}{2}}}{\left(\frac{2\pi d}{p}\right)^{\frac{1}{2d}}} \approx e^{\frac{1}{p}}$$

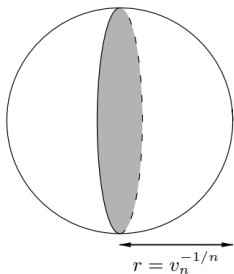

$$r = v_n^{-1/n}$$

Figure 6: Comparing the volume of a ball with that of its central slice

where the second equality is due to Stirling's formula for $\Gamma(1+\frac{d-1}{p})$ and $\Gamma(1+\frac{d}{p})$; the third equality just eliminate the exponential of 2 and $\Gamma(1+\frac{1}{p})$. Thus, we conclude that the slice has volume about $e^{\frac{1}{p}}$ when $d$ is large.

Then, consider the $(d-1)$-dimensional volumes of parallel slices. The slice at distance $x$ from the center is an $(d-1)$-dimensional ball whose radius is $(r^p - x^p)^{\frac{1}{p}}$, so the volume of the smaller slice is about

$$e^{\frac{1}{p}}\left(\frac{(r^p - x^p)^{\frac{1}{p}}}{r}\right)^{d-1} = e^{\frac{1}{p}}\left(1 - \left(\frac{x}{r}\right)^p\right)^{\frac{d-1}{p}}$$

Since $r$ is roughly $\dfrac{d^{\frac{1}{p}}}{2(pe)^{\frac{1}{p}}\Gamma(1+\frac{1}{p})}$, this is about

$$e^{\frac{1}{p}}\left(1-\left(\frac{2x(pe)^{\frac{1}{p}}\Gamma(1+\frac{1}{p})}{d^{\frac{1}{p}}}\right)^p\right)^{\frac{d-1}{p}} = e^{\frac{1}{p}}\left(1-\frac{pe(2x\Gamma(1+\frac{1}{p}))^p}{d}\right)^{\frac{d-1}{p}} \approx \exp\left(\frac{1}{p}-e(2x\Gamma(1+\frac{1}{p}))^p\right)$$

Thus, if we project the mass distribution of the $l_p$ ball of volume 1 onto a single coordinate direction, we get a distribution with density function $f(x) = \exp(\frac{1}{p} - e(2x\Gamma(1+\frac{1}{p}))^p) = \exp\left(\frac{1}{p} - e(\frac{2}{p}\Gamma(\frac{1}{p}))^p x^p\right)$.

Thus, for an $l_p$ ball centered at original point $O$ with volume 1 and approximate radius $\dfrac{d^{\frac{1}{p}}}{2(pe)^{\frac{1}{p}}\Gamma(1+\frac{1}{p})}$, then we can use the integral $2\int_0^s \exp\left(\frac{1}{p} - e(2x\Gamma(1+\frac{1}{p}))^p\right)dx$ to estimate the volume between two parallel slices at the same distance $s$ from the center. Then the volume of $l_p$ ball cap corresponding to the slice at distance $s$ from the center can be approximated by $\frac{1}{2} - \int_0^s \exp\left(\frac{1}{p} - e(2x\Gamma(1+\frac{1}{p}))^p\right)dx$.

Note that the ratio $k$ of slice's distance $d$ from center to radius $r$ is about $s \bigg/ \dfrac{d^{\frac{1}{p}}}{2(pe)^{\frac{1}{p}}\Gamma(1+\frac{1}{p})} = \dfrac{2s(pe)^{\frac{1}{p}}\Gamma(1+\frac{1}{p})}{d^{\frac{1}{p}}}$, i.e. $s = \dfrac{kd^{\frac{1}{p}}}{2(pe)^{\frac{1}{p}}\Gamma(1+\frac{1}{p})}$. Thus, the volume of cap can be represented as

$$\frac{1}{2} - \int_0^{\frac{kd^{\frac{1}{p}}}{2(pe)^{\frac{1}{p}}\Gamma(1+\frac{1}{p})}} \exp\left(\frac{1}{p} - e(2x\Gamma(1+\frac{1}{p}))^p\right)dx$$

which is only related to the ratio $k$. Then, we can conclude that for a $l_p$ ball with radius $r$, when dimension $d$ is large enough and its cap corresponding to the slice at distance $h$ form the center, then the volume ratio of cap to ball is approximately

$$\frac{1}{2} - \int_0^{\frac{sd^{\frac{1}{p}}}{2r(pe)^{\frac{1}{p}}\Gamma(1+\frac{1}{p})}} \exp\left(\frac{1}{p} - e(2x\Gamma(1+\frac{1}{p}))^p\right)dx \tag{151}$$

Thus,

$$\frac{\text{Vol}(\Omega_2)}{\text{Vol}(K)} = \frac{1}{2} - \int_0^{\frac{\epsilon d^{\frac{1}{p}}}{4r(pe)^{\frac{1}{p}}\Gamma(1+\frac{1}{p})}} \exp\left(\frac{1}{p} - e(2x\Gamma(1+\frac{1}{p}))^p\right)dx \tag{152}$$

and therefore

$$TV(\mu, \epsilon e_1 + \mu) = 1 - \frac{\text{Vol}(K \cap (\epsilon e_1 + K))}{\text{Vol}(K)} = 1 - \frac{2\text{Vol}(\Omega_2)}{\text{Vol}(K)}$$

$$= 2 \int_0^{\frac{\epsilon d^{\frac{1}{p}}}{4r(pe)^{\frac{1}{p}}\Gamma(1+\frac{1}{p})}} \exp\left(\frac{1}{p} - e(2x\Gamma(1 + \frac{1}{p}))^p\right) dx$$

$\square$

## J   PROOF OF THEOREM 3.7

*Proof.* Note that $f(x)$ and $g(x)$ are respectively density functions of reference measure $\rho = x + \mu$ and perturbed measure $\nu$ and $q(x)$ is defined as $g(x) - f(x)$. Therefore

$$\mathbb{E}_{X \sim \nu}[\phi(X)] = \int \phi(x)g(x)dx = \int \phi(x)\big(g(x) - f(x)\big)dx + \int \phi(x)f(x)dx = \int \phi(x)q(x)dx + \int \phi(x)f(x)dx$$

$$= \int \phi(x)q(x)dx + \mathbb{E}_{X \sim x+\mu}[\phi(X)]$$

where the first term contains all the uncertainty in one functional variable $q(x)$ and the second term is a constant when sample point $x$, smoothing measure $\mu$ and specification $\phi$ are fixed. And when $\nu \in \mathcal{D}_{x,\delta,p}$ or equivalently $W_p(\nu, x + \mu) \leq \delta$, applying the dual form of $W_p$ distance given in formula 30 and 31, we have

$$W_1(\nu, x + \mu) = \sup_{\varphi \in \mathcal{F}_1} \int \varphi(x)(f - g)(x)dx = \sup_{\varphi \in \mathcal{F}_1} \int \varphi(x)(g - f)(x)dx = \sup_{\varphi \in \mathcal{F}_1} \int \varphi(x)q(x)dx$$

$$= \sup_{||f||_L \leq 1} \int f(x)q(x)dx \leq \delta \tag{153}$$

And when $\nu \in \mathcal{D}_{x,\xi}$ or equivalently $TV(\nu, x + \mu) \leq \xi$, applying lemma A.3 for absolutely continuous measure, we have

$$TV(\nu, x + \mu) = \frac{1}{2} \int |f(x) - g(x)|dx = \frac{1}{2} \int |q(x)|dx \leq \xi \tag{154}$$

It follows that $OPT(\phi, x + \mu, \mathcal{D}_{x,\delta,p} \cap \mathcal{D}_{x,\xi})$ is equivalent to $\min_{\nu \in \mathcal{D}_{x,\delta,p} \cap \mathcal{D}_{x,\xi}} \mathbb{E}[\phi(X)]$ according to the definition and therefore equivalent to optimization problem 23 which is obviously convex according to 153 and 154. $\square$

## K   PROOF OF THEOREM 3.8

Recall the following result proved in the section before

$$\mathbb{E}_{X \sim \nu}[\phi(X)] = \int \phi(x)q(x)dx + \mathbb{E}_{X \sim x+\mu}[\phi(X)] \tag{155}$$

When $\nu \in \mathcal{D}_{x,\delta,p}$ or equivalently $W_p(\nu, x + \mu) \leq \delta$, applying the dual form of $W_p$ distance given in formula 32 and noticing that $\sup_{y \in \text{spt}(\nu) \cup \text{spt}(x+\mu)} ||y||_2 = ||x||_2 + R + \max\{\epsilon, \epsilon d^{\frac{1}{2} - \frac{1}{q}}\} := R^*$, we have

$$\left(\sup_{\varphi \in \text{Lip}(p(2R^*)^{p-1})} \int \varphi(y)(g - f)(y)dy - (p-1)(2R^*)^{p-1}\right)^{\frac{1}{p}} \leq W_p(\nu, x + \mu) \leq \delta \tag{156}$$

or equivalently

$$\sup_{\varphi \in \text{Lip}(p(2R^*)^{p-1})} \int \varphi(y)(g - f)(y)dy = \sup_{||f||_L \leq p(2R^*)^{p-1}} \int f(x)q(x)dx \leq \delta^p + (p-1)(2R^*)^{p-1}$$

$$\tag{157}$$

where $\text{Lip}\big(p(2R^*)^{p-1}\big)$ denotes all maps $f$ from $\mathbb{R}^d$ to $\mathbb{R}$ such that $|f(x) - f(y)| \leq p(2R^*)^{p-1}||x - y||$ for all $x, y \in K$. Note $OPT(\phi, x + \mu, \mathcal{D}_{x,\delta,p} \cap \mathcal{D}_{x,\xi})$ is equivalent to $\min_{\nu \in \mathcal{D}_{x,\delta,p} \cap \mathcal{D}_{x,\xi}} \mathbb{E}[\phi(X)]$ according to the definition and therefore can be relaxed into optimization problem which is obviously convex according to 157 and 154.

## L  PROOF OF THEOREM 3.9

*Proof.* For $0 < p \le 1$, the optimization over $q$ can be solved using Lagrangian duality as follows: we dualize the constraints on $q$ and obtain

$$L(\lambda) = \inf_{||q||_1 \le 2\xi} \left( \int \phi(x)q(x)dx + \mathbb{E}_{X \sim x+\mu}[\phi(X)] + \lambda \left( \sup_{||f||_{L,p} \le 1} \int f(x)q(x)dx - \delta \right) \right)$$

$$= \mathbb{E}_{X \sim x+\mu}[\phi(X)] + \inf_{||q||_1 \le 2\xi} \sup_{||f||_{L,p} \le 1} \left( \int \phi(x)q(x)dx + \lambda \left( \sup_{||f||_L \le 1} \int f(x)q(x)dx - \delta \right) \right)$$

$$= \mathbb{E}_{X \sim x+\mu}[\phi(X)] + \inf_{||q||_1 \le 2\xi} \sup_{||f||_{L,p} \le 1} \int \big( \phi(x) + f(x) \big) q(x)dx - \lambda \delta$$

$$= \mathbb{E}_{X \sim x+\mu}[\phi(X)] + \sup_{||f||_{L,p} \le 1} \inf_{||q||_1 \le 2\xi} \int \big( \phi(x) + f(x) \big) q(x)dx - \lambda \delta$$

$$= \mathbb{E}_{X \sim x+\mu}[\phi(X)] + \sup_{||f||_{L,p} \le 1} \inf_{||q||_1 \le 2\xi} - \int \big| \big( \phi(x) + f(x) \big) q(x) \big| dx - \lambda \delta \tag{158}$$

$$= \mathbb{E}_{X \sim x+\mu}[\phi(X)] - \inf_{||f||_{L,p} \le 1} \sup_{||q||_1 \le 2\xi} \int \big| \big( \phi(x) + f(x) \big) q(x) \big| dx - \lambda \delta$$

$$= \mathbb{E}_{X \sim x+\mu}[\phi(X)] - 2\xi \inf_{||f||_{L,p} \le 1} \big|\big| \phi(x) + f(x) \big|\big|_\infty - \lambda \delta \tag{159}$$

$$= \mathbb{E}_{X \sim x+\mu}[\phi(X)] - 2\xi - \lambda \delta \tag{160}$$

where 158 is due to the choice of $q(x)$ such that $\mathrm{sgn}(q(x)) = \mathrm{sgn}(\phi(x) + f(x))$; 159 is due to Holder inequality when $q = 1, p = \infty$; 160 is due to the fact that $\inf_{||f||_L \le 1} \big|\big| \phi(x) + f(x) \big|\big|_\infty = 1$ since the range of $\phi(x)$ is $\{\pm 1\}$ in applications and $f$ cannot change suddenly when crossing the decision region boundary of $\phi$ due to the Lipschitz constant constraint.

Similarly, for $p > 1$, we have

$$L(\lambda)$$

$$= \inf_{||q||_1 \le 2\xi} \left( \int \phi(x)q(x)dx + \mathbb{E}_{X \sim x+\mu}[\phi(X)] + \lambda \left( \sup_{||f||_L \le p(2R^*)^{p-1}} \int f(x)q(x)dx - \big( \delta^p + (p-1)(2R^*)^{p-1} \big) \right) \right)$$

$$= \mathbb{E}_{X \sim x+\mu}[\phi(X)]$$

$$\quad + \inf_{||q||_1 \le 2\xi} \sup_{||f||_L \le p(2R^*)^{p-1}} \left( \int \phi(x)q(x)dx + \lambda \left( \sup_{||f||_L \le p(2R^*)^{p-1}} \int f(x)q(x)dx - \big( \delta^p + (p-1)(2R^*)^{p-1} \big) \right) \right)$$

$$= \mathbb{E}_{X \sim x+\mu}[\phi(X)] + \inf_{||q||_1 \le 2\xi} \sup_{||f||_L \le p(2R^*)^{p-1}} \int \big( \phi(x) + f(x) \big) q(x)dx - \lambda \big( \delta^p + (p-1)(2R^*)^{p-1} \big)$$

$$= \mathbb{E}_{X \sim x+\mu}[\phi(X)] + \sup_{||f||_L \le p(2R^*)^{p-1}} \inf_{||q||_1 \le 2\xi} \int \big( \phi(x) + f(x) \big) q(x)dx - \lambda \big( \delta^p + (p-1)(2R^*)^{p-1} \big)$$

$$= \mathbb{E}_{X \sim x+\mu}[\phi(X)] + \sup_{||f||_L \le p(2R^*)^{p-1}} \inf_{||q||_1 \le 2\xi} - \int \big| \big( \phi(x) + f(x) \big) q(x) \big| dx - \lambda \big( \delta^p + (p-1)(2R^*)^{p-1} \big)$$

$$= \mathbb{E}_{X \sim x+\mu}[\phi(X)] - \inf_{||f||_L \le p(2R^*)^{p-1}} \sup_{||q||_1 \le 2\xi} \int \big| \big( \phi(x) + f(x) \big) q(x) \big| dx - \lambda \big( \delta^p + (p-1)(2R^*)^{p-1} \big)$$

$$= \mathbb{E}_{X \sim x+\mu}[\phi(X)] - 2\xi \inf_{||f||_L \le p(2R^*)^{p-1}} \big|\big| \phi(x) + f(x) \big|\big|_\infty - \lambda \big( \delta^p + (p-1)(2R^*)^{p-1} \big)$$

$$= \mathbb{E}_{X \sim x+\mu}[\phi(X)] - 2\xi - \lambda \big( \delta^p + (p-1)(2R^*)^{p-1} \big)$$

$\square$

## M    PROOF OF THEOREM 3.10

Recall the certificate by using Hockey-stick divergence provided in table 4 in Dvijotham et al. (2020) as below

$$\epsilon_{\text{HS},\beta} \leq \left[\frac{\beta(\theta_a - \theta_b) - |\beta - 1|}{2}\right]_+$$

When $\beta = 1$, it follows that

$$\epsilon_{\text{HS},1} \leq \left[\frac{\theta_a - \theta_b}{2}\right]_+$$

Besides, recall the relaxation radius using Hockey-stick divergence as below

$$\epsilon_{\text{HS},1} = G\left(\frac{\epsilon}{2\sigma}\right) - G\left(-\frac{\epsilon}{2\sigma}\right) = 2G\left(\frac{\epsilon}{2\sigma}\right) - 1$$

And plug it in above inequality, we have

$$\epsilon_{\text{HS},1} = 2G(\frac{\epsilon}{2\sigma}) - 1 \leq \left[\frac{\theta_a - \theta_b}{2}\right]_+$$

And recall the definition of $\theta_a$ and $\theta_b$, we have

$$\mathbb{E}_{X \sim x + \mu}[\phi(X)] = \theta_a - \theta_b$$

Thus, our certificate $\mathbb{E}_{X \sim x + \mu}[\phi(X)] - 2\left(2G(\frac{\epsilon}{2\sigma}) - 1\right) \geq 0$ is equivalent to

$$2G(\frac{\epsilon}{2\sigma}) - 1 \leq \frac{\theta_a - \theta_b}{2}$$

Thus, the equivalence relation holds.

