# OpenReview forum: "Certified Adversarial Robustness Under the Bounded Support Set"
_ICLR.cc/2022/Conference — ICLR 2022 Submitted_

### Official Review · Reviewer_neZn · 2021-10-29

**Correctness:** 4
**Technical Novelty And Significance:** 3
**Empirical Novelty And Significance:** 2
**Recommendation:** 3
**Confidence:** 3

**Main Review:**

**Strengths:**
The paper is well organized and clearly explains the approach considered. The work is technically impressive and involves some fairly complicated derivations. The proposed approach is significantly more efficient that Dvijotham's approach. There are several results here, though sadly, many of them are negative in nature.

**Weaknesses:**
A primary downfall of this work is the motivation. It is unclear under what circumstances the designer of a robustness certificate does not have control of the smoothing distribution. The standard and accepted setting is that we desire a classifier that is robust to a specified threat model, and we are allowed to choose whichever smoothing distribution we like to achieve this goal. It is not well motivated why the opposite question should be asked: "how can one apply the full-information robust certification approach of Dvijotham when the smoothing distribution is fixed a priori?". Without such a motivation, the theorems read like a laundry list of smoothing distributions for which we can prove properties, instead of a series of hard-fought and significant contributions.

Additionally, the experiments section is unsatisfying. While it is true that no other works consider uniform distributions when smoothing, no mention is paid to the comparison of Lp-robustness certificates against competing approaches that use the same base classifier (but possibly different smoothing distributions). Is this because the uniform smoothing distribution vastly underperforms when compared to these settings?

**Other notes:**
- I've never seen "total variation distance" referred to as "total variance distance." Maybe this is valid, but it is certainly nonstandard
- The authors should probably cite Yang et al which describes optimal smoothing distributions to attain Lp robustness.


**Summary Of The Paper:**

The authors provide robustness certification results for smoothed classifiers. Specifically, the considered setting follows that of Dvijotham (2020) except the base smoothing distribution is specified to have bounded support, and the relaxation follows from an intersection of Wasserstein and TV balls. Under this setting, much of the technical machinery describes which Wasserstein and TV balls are needed to cover these bounded-support smoothing distributions.

**Summary Of The Review:**

While technically very sound, this work faces serious problems regarding motivation and provides several negative-seeming results regarding smoothing using uniform distributions. I do not recommend for acceptance, but could raise my score to borderline if a convincing argument of motivation is provided.

---

> ### Author Response · Authors · 2021-11-21
> **Clarification on the weakness the reviewer mentioned**
>
> We would first like to thank the reviewer for the time reviewing our work and valuable comments. In the following, we will provide a step-by-step response to the comments of the reviewer.
>
> $\bullet$ As for the motivation, the initial purpose of our work is to generalize Dvijotham's f-divergence-based framework to a W-distance-based framework, because we have noticed that f-divergence is only able to analyze smoothing distribution with support set $R^{d}$ due to its definition as $D_f(p||q)=\int q(x)f(\frac{p(x)}{q(x)})dx$ which is related to the likelihood ratio $\frac{p(x)}{q(x)}$. The reason why is unable to analyze smoothing measure with bounded support set is that when $supp(p(x)):=K$ is bounded and $q(x)=p(x-v),supp(q(x))=v+K$ where $v$ is a displacement vector and when $x\in K\setminus(v+K)$ it follows that $\frac{p(x)}{q(x)}=\frac{p(x)}{0}$ and therefore the likelihood ratio and f-divergence are undefined. By generalizing this framework using W-distance, it is able to analyze smoothing distribution with bounded support set due to the dual form of $W_1$ distance $W_1(\mu,\nu)=\sup_{f_L\leq1}E_{x\sim\mu}[f(x)]-E_{x\sim\nu}[f(x)]=\sup_{f_L\leq1}E_{x\sim\mu-\nu}[f(x)]$ and the definition of TV distance $TV(\mu,\nu)=\sup_A$|$\mu(A)-\nu(A)$|, which have nothing to do with the likelihood ratio and are only related to the difference density function. And following this thread, we propose a certification formula based on W distance and TV distance which can be used to analyze the performance of smoothing measure with bounded support set. It worth mentioning that when applying our framework and Dvijotham's framework to Gaussian smoothing measure, we can obtain the same certification radius as stated in section 3.4. And our framework can also analyze smoothing measure with bounded support set. And all the theorems listed together contribute to the certification formulas listed in table 1 and the organization of these theorems has been discussed in the reply to the Reviewer 55UR. And the negative performance of uniform smoothing measure with bounded support set is just a natural and consequent conclusion when applying our W-distance and TV-distance based framework to uniform smoothing distribution, although the performance looks negative due to the incapability of uniform smoothing measure in nature. And I have to stress again that our contribution is the capability of obtaining certification formula for smoothing measures with bounded support sets, which is definitely unlimited to uniform smoothing measures and include the truncated and normalized version of any probability measure such as Gaussian and Laplace, etc. And we wish this make sense for the reviewer for the motivation.

---

### Official Review · Reviewer_55UR · 2021-11-03

**Correctness:** 3
**Technical Novelty And Significance:** 3
**Empirical Novelty And Significance:** 2
**Recommendation:** 5
**Confidence:** 3

**Main Review:**

**Strengths**

The theoretical analysis is comprehensive and sound.

**Weakness**
1. There is a huge gap between the theoretical analysis and experimental results. Around 10 Theorems proposed in the paper but I'm not sure whether all of them are necessary yet increase the reading difficulty.

2. Algorithm 1 is unnecessary since it's almost the same as existing randomized smoothing based algorithms.

3. The certified bounds in experimental parts like in Figure 2 and 3 are very trivial. I understand that the results are given by a natural trained ResNet-110, but the certified accuracy on robust trained models is also significant but missing at all in the paper. Moreover, it looks like the proposed method does support robust training but there are no experimental results shown here.

4. The paper claimed multiple times on the efficiency of the proposed compared with Dvijotham et al. (2020) but considering Dvijotham et al. (2020) has experimental results on the ImageNet dataset, it would be more convincing to add experiments on such large-scale datasets. Also, consider most conventional randomized smoothing based methods have constant computation time, so I think the author should turn down the statement "an improvement upon the state-of-the-art methods in terms of the computation time" since actually there is only one other method (Dvijotham et al. (2020)) you mentioned here.

5. No comparison experiment with conventional randomized smoothing based methods and a very related work is missing:
 Yang, Greg, et al. "Randomized smoothing of all shapes and sizes." International Conference on Machine Learning. PMLR, 2020.


**Summary Of The Paper:**

The paper proposed a framework based on Wasserstein distance and total variance distance relaxation as well as Lagrange duality to deal with the analysis of bounded support set smoothing measures. The experimental results show the relative incapability of bounded support set smoothing measures compared with Gaussian smoothing measures.

**Summary Of The Review:**

I think the current version of the paper is not well-ready to ICLR quality, there are multiple missing parts in related work and experiments sections.

---

> ### Author Response · Authors · 2021-11-20
> **Clarification on the weakness the reviewer mentioned**
>
> Thanks for the reviewer's valuable comments. Here we would like to clarify some weakness the reviewer has mentioned.
>
> $\bullet$ For point 1, we omit the experiment of comparing our method and the conventional smoothing methods after we find out that our method can achieve exactly the same results as the method in Dvijotham et al. (2020). It's of little use repeating such comparison since it's proved to be the same in certain settings (section 3.4), and we assume they will perform similarly with other reasonable parameter settings. So instead, we focus on the internal comparisons using different smoothing distributions. As for the theorems proposed, they are all necessary part of our theoretical framework and some of them have no corresponding experiment because they theoretically omit the probable good performance. To make everything clear, I will introduce the organization of all these theorems below:
>
> $\quad\bullet$ First, our framework consists of three parts:  computing the Wasserstein distance relaxation measure set,
> computing the total variance distance relaxation measure set, and computing the Lagrange function
> as well as dual problem of the relaxed optimization problem as the paragraph before section 3.1 in our paper stated.
>
> $\quad\bullet$ Theorem 3.1 proves that perturbed probability set is bounded w.r.t. Wasserstein distance measure, which is essential for the relaxed optimization problem.
>
> $\quad\bullet$ Theorem 3.2 gives TV distance relaxation for Gaussian which results in theorem 3.10 in section 3.4 about the equivalence of both framework when applying to Gaussian distribution.
>
> $\quad\bullet$ Theorem 3.3 theoretically gives bad performance of $l_1$ ball support set for $l_2,l_\infty$ adversary and therefore we did no experiments corresponding to this case.
>
> $\quad\bullet$ Theorem 3.4, 3.5, 3.7, 3.8, 3.9 all contribute to the certification formula in our experiments.
>
> $\quad\bullet$ Theorem 3.6 theoretically gives poor performance for uniform measure with general $l_p$ norm ball support set.
>
> $\bullet$ For point 2, our method focuses on the more efficient ways of calculating certification bounds and doesn't change much of the general procedure, but we still think such pseudo code demonstration is needed for readers' easier comprehension.
>
> $\bullet$ For point 3, testing out the performance of models with robust training may yield better certification results, but such improvement applies to other models as well. We think it doesn't add up much to our point of efficiency improvement and the use of normal distribution, and thus is not relevant enough to be included in our paper.
>
> $\bullet$ For point 4, since ours and the method in Dvijotham (2020) take pre-set amount of samples or iterations for each data point, their computational time should be proportional to the dataset size with respective growth rates, and we think the CIFAR-10 10000 image test set is already sufficient to illustrate such difference in the growth rates of time cost.
>
> $\bullet$ For point 5, in the paper Yang, Greg, et al. "Randomized smoothing of all shapes and sizes." the reviewer mentioned, their experiments mainly focus on comparing relative performance between Gaussian, Laplace, Exponential, PowerLaw and Uniform smoothing distribution with respect to certain norm under certain adversary norm, which is demonstrated in figure 2 and 3 in their paper and leads to conclusions that which one among these particular kinds of distribution performs better respectively for $l_1,l_2,l_\infty$ adversary. However, our paper mainly focus on uniform smoothing distribution with support set with ball of different $l_q$ norm and different ball radius $r=0.025,0.05,0.1$. For $l_1$ adversary, we didn't do experiment because in theorem 3.5 and table 1 we have shown that for uniform distribution with $l_\infty$ support set, the TV distance relaxation radius for $l_1$ adversary has nothing to do with the data dimension $d$ and the certification formula $E_{x\sim X+\mu}[\phi(x)]-\frac{\epsilon}{r}$ theoretically turns out to be fine. This theoretical result meets the conclusion in Yang's paper that the Wulff Crystal for the $l_1$ ball is a cube and the experiment result in figure 1 in their paper that uniform smoothing distribution performs best among the distribution family they list, although our paper and their paper in fact use totally different deduction method. (Their paper mainly uses NP lemma, while our paper is mainly based on convex relaxtion technique following Dvijotham's work.) And for $l_\infty$ adversary, their paper proves that the Wulff Crystal is the zonotope of vectors $\\{\pm1\\}^{d}$, which is a highly complex polytope hard to sample from and related to many open problems in polytope theory. Owing to these considerations, we only did experiments on $l_2$ adversary with uniform smoothing distribution of $l_2$ ball and $l_\infty$ ball support set and made no comparison experiment because of different focus as stated above.

---

### Official Review · Reviewer_hLLe · 2021-11-03

**Correctness:** 4
**Technical Novelty And Significance:** 2
**Empirical Novelty And Significance:** Not applicable
**Recommendation:** 3
**Confidence:** 4

**Main Review:**

The idea of this paper is to use analytic tools to enable randomized smoothing with bounded smoothing distributions, mainly uniform distribution within a symmetric Lp ball, similar to Gaussian and Laplacian distribution in previous works.

While the motivation is good, the empirical performance of the uniformly smoothed classifiers s are much worse than the Gaussian smoothed classifier. It’s worth mentioning that this should be expected as previous work has drawn conclusions that most i.i.d smoothing distributions suffer from the curse of dimensionality [1]. More importantly, in Section 5 of [1], it has been shown that uniform distributions suffer from the curse of dimensionality, in the sense that the certifiable radius is upper bounded by a factor of $(1/d^{1-1/p}$, where $d$ Is the dimensions and $p$ Is for the corresponding $L_p$ measure.

In some sense, [1] has shown that uniform distributions can be used for randomized smoothing, and the fact that it’s not a good choice.

[1] Kumar, Aounon, et al. "Curse of dimensionality on randomized smoothing for certifiable robustness." International Conference on Machine Learning. PMLR, 2020.

**Summary Of The Paper:**

In this paper, the authors aim to improve previous works on randomized smoothing by extending the analysis to allow smoothing measures with bounded support.

**Summary Of The Review:**

While the analytic efforts in this paper are both deep and comprehensive, the problem the authors try to solve has been studied with a quite decisive and general conclusion. Considering the overlaps, I cannot recommend accepting this paper.

---

> ### Author Response · Authors · 2021-11-18
> **Several clarification about the differences between our paper and [1]**
>
> We would first like to thank the reviewer for the time reviewing our work. In the following, we will provide a step-by-step response to the comments of the reviewer.
>
> In section 5 of paper [1] the reviewer mentioned, the author only discussed the theoretical performance related to uniform smoothing distribution of $l_1$ norm ball and $l_{\infty}$ norm ball support set. And the parameter $p$ in [1] is corresponding to the norm of adversarial perturbation rather than the norm of support set of uniform smoothing measure.
>
> However, in theorem 3.6 in our paper, the definition of parameter $p$ is corresponding to the norm of support set of uniform smoothing measure which is a totally different concept from that of paper [1], while parameter $q$ is the same concept with parameter $p$ in paper [1]. In regard of this, our work actually consider a more general case than that of [1] by taking the performance of uniform smoothing distribution of $l_p$ norm ball support set with regard to $l_q$ norm adversarial perturbation into consideration. After making the definition of parameter $p$ clear, we can better understand the organization of section 3.2.2:
>
> $\bullet$ In theorem 3.3, we discussed about the $p=1$ case.
>
> $\bullet$ In theorem 3.4, we discussed about the $p=2$ case.
>
> $\bullet$ In theorem 3.5, we discussed about the $p=\infty$ case.
>
> $\bullet$ In theorem 3.6, we discussed about the general $p$ case.
>
> And paper [1] only discussed about the special case when $p=1$ as well as $p=\infty$. Moreover, as for the overlapping case of $p=1,\infty$, the theoretical deriviation methods our paper and paper [1] used are quite different and therefore result in distinguished results. The differences are listed as below.
>
> $\bullet$ The certified radius paper [1] provided for the $l_{\infty}$ case ($r_p^*<\frac{2b}{d^{1-\frac{1}{p}}}$) and $l_{1}$ case ($r_p^*<\frac{2b}{d}$) are independent of the probability of top two classes $p_1(x)$ and $p_2(x)$. The reason is that in their proof of theorem 4 and 5, they considered the radius working for all possible classifiers and therefore was able to take a special classifier $g$ to facilitate their proof, just as stated in the introduction section in our paper (several lines in the bottom of page1). In light of this, although they provided several hardness results with respect to uniform smoothing measure, this result might be overtight because they took into account classifiers that will never appear in real-world applications, which also results in the incapability to derive a certification formula for later experiment. Our work does not apply
>  such relaxation but provides a certification formula related to certain classifier $E[\phi(x)]$ (equivalent to $p_1(x)-p_2(x)$ for binary classifier) and is therefore able to show the poor performance of uniform smoothing measure both theoretically and experimentally.

---

### Decision · Program_Chairs · 2022-01-20

**Decision:**

Reject

**Comment:**

Authors study robustness properties of arbitrary smoothing measures with bounded support using Wasserstein distance and total variation distance. Reviewers pointed out several weaknesses about this work. In particular, they mentioned the paper is not well-organized, comparison with prior work is lacking, the conclusion of the theoretical analysis is not novel and the experiments are not comprehensive. I suggest authors to take these comments into account in improving their work.